# Two mechanisms regulate directional cell growth in *Arabidopsis* lateral roots

**Charlotte Kirchhelle[1]\*, Daniel Garcia-Gonzalez[2,3], Niloufer G Irani[1], Antoine Jérusalem[2], Ian Moore[1]†**

[1]Department of Plant Sciences, University of Oxford, Oxford, United Kingdom; [2]Department of Engineering Science, University of Oxford, Oxford, United Kingdom; [3]Department of Continuum Mechanics and Structural Analysis, University Carlos III of Madrid, Madrid, Spain

**Abstract** Morphogenesis in plants depends critically on directional (anisotropic) growth. This occurs principally perpendicular to the net orientation of cellulose microfibrils (CMFs), which is in turn controlled by cortical microtubules (CMTs). In young lateral roots of *Arabidopsis thaliana*, growth anisotropy also depends on RAB-A5c, a plant-specific small GTPase that specifies a membrane trafficking pathway to the geometric edges of cells. Here we investigate the functional relationship between structural anisotropy at faces and RAB-A5c activity at edges during lateral root development. We show that surprisingly, inhibition of RAB-A5c function is associated with increased CMT/CMF anisotropy. We present genetic, pharmacological, and modelling evidence that this increase in CMT/CMF anisotropy partially compensates for loss of an independent RAB-A5c-mediated mechanism that maintains anisotropic growth in meristematic cells. We show that RAB-A5c associates with CMTs at cell edges, indicating that CMTs act as an integration point for both mechanisms controlling cellular growth anisotropy in lateral roots.

DOI: https://doi.org/10.7554/eLife.47988.001

**\*For correspondence:**
charlotte.kirchhelle@plants.ox.ac.uk

†Deceased

**Competing interests:** The authors declare that no competing interests exist.

## Introduction

Plants display an astonishing morphological diversity both at the organ and the cellular scale. As plant cells are encased by a rigid cell wall and cannot migrate from their position in the tissue, this diversity in shape depends on the plant's ability to precisely control directional cell growth. Plant cells grow through plastic deformation of the cell wall, a process driven by the cell's undirected internal turgor pressure (*Lockhart, 1965*). Directional growth therefore depends on differential extensibility of cell walls at different cell faces, which in turn is achieved through local modifications of cell wall mechanical and structural properties.

Plant cell walls consist of three main polysaccharides (cellulose, hemicelluloses, and pectins) as well as a small fraction of enzymatic and structural proteins (*Cosgrove, 2014*). Cellulose microfibrils (CMFs) have a long been seen as the most important cell wall component in terms of structural anisotropy (*Green, 1962*). With a Young's modulus in fibre direction of ~130 GPa, they are much stiffer than other cell wall components (*Burgert, 2006*), and therefore constrain cell wall extensibility in fibre direction. Consequently, the leading paradigm for directional growth control predicts directional growth is controlled through oriented deposition of CMFs in the cell wall, with maximum expansion perpendicular to the net orientation of CMFs (*Green, 1962*; *Green, 1980*). CMF orientation is determined during deposition by cortical microtubule (CMT) arrays, which guide cellulose synthase complex (CSC) trajectories in the plasma membrane (*Bringmann et al., 2012*; *Li et al., 2012*; *Paredez et al., 2006*). However, the CMT/CMF paradigm alone is insufficient to explain directional plant growth in various cases (*Baskin et al., 1999*; *Taiz and Métraux, 1979*). In an intriguing classic example, pharmacological or genetic perturbation of transverse CMT patterns in elongating

**eLife digest** A fundamental challenge in biology is to understand how plants, animals and other multicellular organisms make organs of various shapes and sizes. Plant cells are surrounded by rigid walls that fix them into their position relative to the surrounding cells. Therefore, neighbouring cells have to precisely coordinate how fast and in which directions they grow to form organs. In roots, for example, cells need to primarily grow in a longitudinal direction (along the length of the root), rather than radially, in order for the root to maintain its cylinder shape.

Plant cell walls are made of fibres known as cellulose microfibrils, which are embedded within a matrix of other cell wall material. These fibres are much stiffer than the cell wall matrix, and their orientation controls in which direction a cell will grow. When microfibrils are lying parallel to each other, they limit cell growth in the direction they are orientated, causing cells to grow at right angles to the microfibrils. However, the way microfibrils are angled does not always fully expain the direction in which a cell will grow: for example, young cells in lateral roots (roots that branch out from the primary root) mainly grow in a longitudinal direction although their microfibrils are randomly arranged. This suggests other mechanisms are also involved in controlling in which direction a cell will grow.

In order to grow, cells transport cell wall material and other cargo to the cell surface through different transport routes. A protein known as RAB-A5c controls a pathway directed to the edges of cells where two faces meet. It is not clear what exactly is transported in this edge-directed pathway, but root cells no longer grow in a longitudinal direction when RAB-A5c is inhibited. Previous studies have found that the edges of cells help to arrange cellulose microfibrils in the cell wall, but it is not clear whether RAB-A5c controls cell growth through this or another process. To address this question, Kirchhelle et al. used several different approaches to study the role RAB-A5c plays in how lateral roots grow in the small weed *Arabidopsis thaliana*.

The experiments found that RAB-A5c controlled which direction cells in the lateral roots grew independently of cellulose microfibril orientation. Inhibiting RAB-A5c activity disturbed longitudinal growth, but unexpectedly increased the proportion of cellulose microfibrils that were arranged at right angles to the root's main axis. Kirchhelle et al. propose that RAB-A5c controls growth direction by changing the cell wall's mechanical properties at the edge where two cell faces meet. Cells can partially compensate for loss of this mechanism by increasing the number of microfibrils that are deposited in a parallel arrangement.

These findings demonstrate that there are two separate mechanisms that control in which direction root cells grow. All land plants have versions of the RAB-A5c protein, suggesting that the new mechanism revealed in this work may also be found in other plants. Understanding how plant cells control growth may benefit agriculture in the future, for example, by providing new targets to develop herbicides against weeds.

DOI: https://doi.org/10.7554/eLife.47988.002

---

*Arabidopsis* roots caused cell swelling which *preceded* changes in cellulose microfibril orientation (*Sugimoto et al., 2003*; *Whittington et al., 2001*; *Wiedemeier et al., 2002*), indicating that (1) anisotropic microfibril orientation is not always sufficient to confer anisotropic growth, and (2) microtubules may control growth anisotropy through mechanisms independent of CMF orientation. In recent years, models of cell wall structure and mechanics have undergone major revisions, incorporating new insights into additional mechanisms of directional growth control (*Cosgrove, 2014*). In particular, the methyl-esterification status of pectins has been attributed a much more important role in determining the structural and mechanical properties of the cell wall (*Wolf and Greiner, 2012*). In the context of anisotropic growth control, differential demethyl-esterification at longitudinal and transverse cell faces in the hypocotyl was associated with a shift from isotropic to anisotropic growth and preceded changes in CMF orientation (*Peaucelle et al., 2015*).

In addition to cell faces, the geometric edges of cells (where two faces meet) have recently emerged as a spatial domain with importance for anisotropic growth control (*Ambrose et al., 2011*; *Kirchhelle et al., 2016*). The plant-specific small Rab GTPase RAB-A5c is an endomembrane trafficking regulator that specifies a putatively exocytic membrane trafficking pathway to geometric edges

in organ primordia (*Rutherford and Moore, 2002*; *Kirchhelle et al., 2016*). Its inhibition caused a shift in cell growth direction from anisotropic to near-isotropic without a change in overall growth rates or default endomembrane trafficking, indicating a requirement for this pathway in directional growth control (*Kirchhelle et al., 2016*). Cells' geometric edges have been identified as important organisational domains for CMTs (*Ambrose et al., 2011*; *Ambrose and Wasteneys, 2011*; *Gunning et al., 1978*). Cell-edge geometry can influence CMTs at the faces as sharp transverse edges present a physical barrier to CMTs, leading to 'edge catastrophe' of longitudinally oriented CMTs encountering such edges and subsequently, transverse CMT arrays (*Ambrose et al., 2011*). The microtubule-associated protein CLASP accumulates at geometric edges of some cells and enables CMTs to overcome these barriers (*Ambrose et al., 2011*). Furthermore, cell edges have been identified as sites of microtubule nucleation in different plant species through accumulation of γ-tubulin complex components (*Ambrose and Wasteneys, 2011*; *Gunning et al., 1978*; *Gunning, 1980*), further contributing to their role as cell-level CMT organisers. However, we have previously proposed the role of RAB-A5c at cell edges may be independent of CLASP-mediated CMT organisation, as only a minor fraction of CLASP and RAB-A5c colocalised at cell edges, and localisation of RAB-A5c to cell edges was independent of CLASP (*Kirchhelle et al., 2016*). Instead, we proposed RAB-A5c may act through locally changing cell wall properties at cell edges, which was supported by a 2D Finite Element (FE) linear elastic model in which reduction of cell wall stiffness edges caused cell swelling.

Here, we use a combination of experimental and computational techniques to investigate the functional relationship between RAB-A5c-mediated trafficking and CMT organisation at cell edges in the context of directional growth control. Specifically, we set out to test whether RAB-A5c activity affects growth anisotropy though changes in CMT organisation, following the classic paradigm for directional growth control, or through an independent mechanism as we have hypothesised before. Combining experimental and computational approaches, we demonstrate that in young lateral roots, both re-organisation of CMTs into anisotropic arrays and RAB-A5c-mediated edge-directed trafficking contribute to anisotropic growth. We provide evidence that these pathways act independently and at different developmental stages, but can partially complement each other. We furthermore demonstrate that RAB-A5c associates with CMTs at cell edges and changes in CMT array organisation affect the RAB-A5c localisation pattern, suggesting CMTs act as an integration point of both mechanisms controlling growth anisotropy in lateral roots: cellulose anisotropy at faces, and RAB-A5c activity at cell edges.

## Results

### CMT and CMF anisotropy *increase* when RAB-A5c is inhibited

To test whether RAB-A5c function at cell edges controlled directional growth through CMT organisation, we introduced the microtubule marker pUBQ1::RFP:TUB6 (*Ambrose et al., 2011*) into the *RPS5a > Dex > RAB-A5c[N125I] pUBQ10::YFP:NPSN12* background (*Kirchhelle et al., 2016*). In these lines, RAB-A5c function can be conditionally disrupted through expression of a dexamethasone (Dex)-inducible, dominant-negative protein variant RAB-A5c[N125I]. As this protein variant is predicted to act through titration of interacting factors, this strategy can overcome redundancy amongst different members of the same gene family as well as allowing temporal and dosage control (*Batoko et al., 2000*; *Jones et al., 1995*; *Olkkonen and Stenmark, 1997*; *Pinheiro et al., 2009*; *Schmitt et al., 1986*). Previous work in these lines revealed gross morphological defects in lateral roots within 48 hr – 72 hr after transfer to Dex-containing medium, with a dramatic shift from the normally highly anisotropic cellular growth to almost fully isotropic growth (*Kirchhelle et al., 2016*).

We induced expression of *RPS5a > Dex > RAB-A5c[N125I]* for 72 hr and quantified mean CMT array anisotropy and orientation. Surprisingly, CMT array anisotropy at the outer periclinal cell face in meristematic lateral root cells was significantly increased in the presence of RAB-A5c[N125I] despite the loss of growth anisotropy (*Figure 1A–C*, *Figure 1—figure supplement 1A,B*). Furthermore, while the CMT arrays of individual cells were essentially randomly oriented relative to the longitudinal axis in wild-type roots, in the presence of RAB-A5c[N125I] they were highly transverse, resulting in a prominent supra-cellular pattern (*Figure 1A,B,D*). This change in mean CMT array orientation occurred within 24 hr of induction with both saturating and sub-saturating concentrations

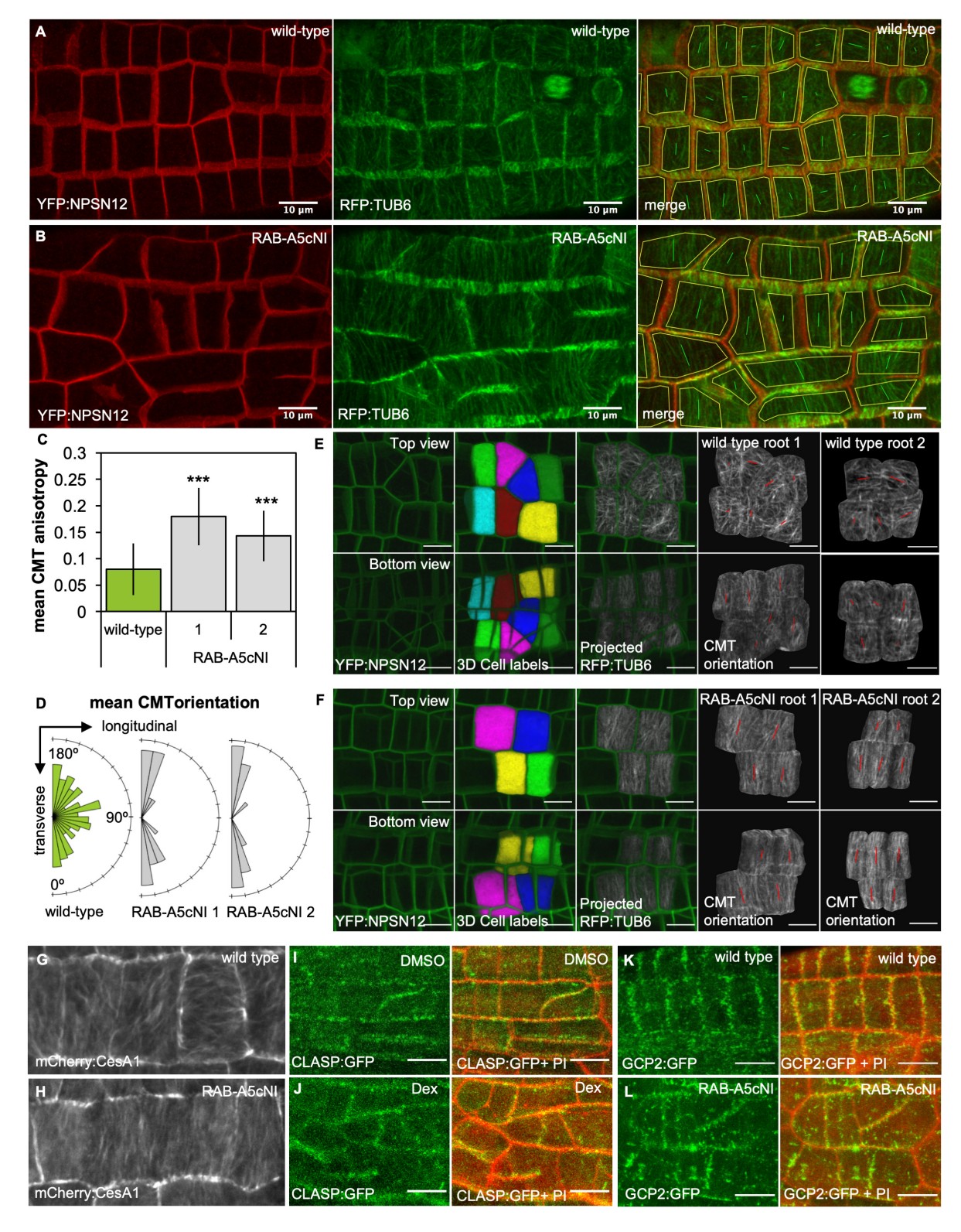

**Figure 1.** Inhibition of RAB-A5c function caused increased anisotropy of the CMT array in meristematic cells. (**A,B**) Maximum intensity projections of confocal stacks of lateral roots co-expressing YFP:NPSN12 (top) and RFP:TUB6 (middle) either in wild-type (**A**) or in a *RPS5a > Dex > RAB-A5c[N125I]* line (**B**) three days after seedlings were transferred to agar plates containing 20 μM Dex. Bottom image shows merge of both channels as well as manually drawn cell outlines (yellow) and vectors indicating mean orientation and degree of anisotropy of the CMT array in each cell as measured with

*Figure 1 continued on next page*

*Figure 1 continued*

FibrilTool (*Boudaoud et al., 2014*). (C) Plot showing anisotropy of CMTs in meristematic cells from lateral roots like those shown in (A) for wild-type (n = 114 cells) and *RPS5a > Dex > RAB-A5c[N125I]* line 1 (n = 43 cells) and line 2 (n = 41 cells). Note 0 corresponds to a fully isotropic array, one to a completely parallel (anisotropic) array. Mean CMT array anisotropy was significantly increased in the presence of RAB-A5c[N125I] (Welch's t-test: p<0.001 (***)). Error bars are SD. (D) Rose diagrams showing mean orientation of the CMT array in cells used in (C) relative to the longitudinal and transverse axes of the lateral root. (E,F) CMT orientation at inner vs. outer periclinal cell walls in wild-type (E) and *RPS5a > Dex > RAB-A5c[N125I]* (F) roots. Epidermal meristematic cells expressing YFP:NPSN12 were segmented in 3D using MorphoGraphX (*Barbier de Reuille et al., 2015*), co-expressed RFP:TUB6 was projected onto the 3D cell mesh, and mean CMT orientation at inner and outer periclinal faces was quantified using with FibrilTool (*Boudaoud et al., 2014*). Seedlings were imaged 24 hr after transfer to agar plates containing 20 µm Dex. (G,H) Average time projections of maximum intensity projections of confocal stacks of lateral roots expressing mCherry:CESA1 in a wild-type (G) or *RPS5a > Dex > RAB-A5c[N125I]* background (H). Seedlings were imaged 24 hr after transfer to agar plates containing 20 µm Dex. Stacks were acquired in 10 s intervals, each time average projection is based on 60 stacks (corresponding to 10 min total imaging time). (I,J) Maximum intensity projections of confocal stacks expressing CLASP:GFP in the *RPS5a > Dex > RAB-A5c[N125I]* background showing lateral roots 3 days after transfer to agar plates containing 0.1% DMSO (I) or 20 µM Dex (J). Cell walls were stained with Propidium Iodide (PI). (K,L) Maximum intensity projections of confocal stacks expressing GCP2:3xGFP in a wild-type (K) and *RPS5a > Dex > RAB-A5c[N125I]* (L) background showing lateral roots 3 days after transfer to agar plates containing 20 µM Dex. Scale bars 10 µm. Cell walls were stained with Propidium Iodide (PI). All scale bars 10 µm.

DOI: https://doi.org/10.7554/eLife.47988.003

The following source data and figure supplements are available for figure 1:

**Source data 1.** Quantification of CMT anisotropy and mean orientation in the absence and presence of RAB-A5c[N125I].
DOI: https://doi.org/10.7554/eLife.47988.008

**Figure supplement 1.** CMTs in the absence and presence of RAB-A5cNI.
DOI: https://doi.org/10.7554/eLife.47988.004

**Figure supplement 2.** CMT rearrangement occurs during early stages of RAB-A5c[N125I] induction.
DOI: https://doi.org/10.7554/eLife.47988.005

**Figure supplement 2—source data 1.** Quantification of CMT anisotropy in epidermal lateral root cells during early stages of RAB-A5c[N125I] induction.
DOI: https://doi.org/10.7554/eLife.47988.006

**Figure supplement 3.** GCP2 and RAB-A5c localisation in lateral roots.
DOI: https://doi.org/10.7554/eLife.47988.007

**Figure supplement 3—source data 1.** Quantification of colocalisation between YFP:RAB-A5c and GCP2:GFP at cell edges.
DOI: https://doi.org/10.7554/eLife.47988.012

of Dex and preceded obvious changes in cell geometry (*Figure 1—figure supplement 2*). Since CMT arrays can differ at outer and inner faces in some cell types (*Crowell et al., 2011*; *Panteris et al., 2013*), we investigated whether CMT array organisation was similarly affected at the inner periclinal face (at the L1/L2 interface). The inner periclinal face can have a complex surface topology depending on its neighbouring cortical cells, so we employed an image analysis strategy involving 3D-segmentation of cells based on the plasma membrane marker YFP:NPSN12, and subsequent projection a co-expressed RFP:TUB6 marker onto the 3D cell surface (*Figure 1 E,F*; *Figure 1—figure supplement 1C,D*). Using this technique, we found that in wild type roots, CMT arrays at the L1/L2 interface had a relatively low anisotropy although they appeared to be oriented in transverse orientation more often than those at the outer epidermal face. In the presence of RAB-A5c[N125I], we observed an increase of CMT anisotropy and transversely oriented arrays at the L1/L2 interface similar to our observations at the outer face. We also introduced the cellulose synthase marker pCESA1::mCherry:CESA1 (*Vain et al., 2014*) into the *RPS5a > Dex > RAB-A5c[N125I]* background and imaged cellulose synthase trajectories 24 hr and 48 hr after induction of RAB-A5c[N125I] (*Figure 1G,F*; *Figure 1—figure supplement 3A,B*). While trajectories were variable in their orientation in wild-type meristematic cells, they were predominantly transverse in the presence of RAB-A5c [N125I], following the pattern observed for CMTs. This indicated that the change in CMT orientation indeed led to an increase in CMF anisotropy, and consequently, cell wall anisotropy.

The surprising increase in CMT and CMF anisotropy appears to contradict the common assumption that anisotropic CMT and CMF arrays are associated with anisotropic growth – however, similarly transverse anisotropic arrays have been described in the *clasp-1* mutant, which also has shorter and more swollen cells than wild-type plants (*Ambrose et al., 2011*; *Ambrose et al., 2007*). We therefore tested if RAB-A5c function was required for localisation of CLASP at cell edges, but found CLASP:GFP still localised to edges in *RPS5a > Dex > RAB-A5c[N125I]* lines 72 hr after transfer to

Dex (*Figure 1I,J*), indicating edge-localisation of CLASP does not depend on RAB-A5c. We also considered that RAB-A5c might act through γ-tubulin complex protein 2 (GCP2), which can act as a CMT array organiser through nucleation of new MTs at transverse cell edges in primary roots (*Ambrose and Wasteneys, 2011*). GCP2:GFP localised to both longitudinal and transverse cell edges in lateral roots (*Figure 1—figure supplement 3C*) and partially overlapped with YFP:RAB-A5c (*Figure 1—figure supplement 3D–F*). GCP2:GFP was enriched approximately two-fold at transverse compared to longitudinal cell edges in lateral roots, so a loss of the MT nucleating activity of GCP2 from these edges could conceivably result in the more transverse CMT array observed in the presence of RAB-A5c[N125I]. However, GCP2:GFP still localised to transverse cell edges in *RPS5a > Dex > RAB-A5c[N125I]* lines 72 hr after transfer to Dex (*Figure 1K,L*). Therefore, the observed effect of RAB-A5c[N125I] on CMT array organisation is likely not mediated through either CLASP or GCP2.

## Increased cell wall anisotropy can counteract edge-mediated cell swelling *in silico*

We previously proposed that RAB-A5c may act through locally changing cell wall mechanical properties at edges. In this context, the observed increase of CMT and CMF anisotropy reported above can be interpreted as a secondary, compensatory effect counteracting the loss of RAB-A5c activity. We used a computational modelling approach to test whether increased CMF anisotropy at cell faces could counteract cell swelling caused by local reduction in cell wall stiffness at the cell edges. The model follows a previous approach for axon contribution to the mechanical behaviour of brain white matter (*Garcia-Gonzalez et al., 2018*), allowing us to independently define the CMF and matrix responses. CMF orientation and degree of anisotropy were introduced as mechanical features of the constitutive model, and the matrix was defined as a hyperelastic isotropic material. We also introduced a term to define cross-linking of CMFs, which we considered to be isotropic and to represent both direct cellulose-cellulose interactions and interactions mediated by matrix components in accordance with the mechanical hotspot theory (*Cosgrove, 2014*). Our constitutive model requires the identification of seven parameters: the bulk modulus $K$; the shear moduli of the cell wall matrix $\mu_m$, cellulose microfibrils $\mu_f$, and microfibril cross-linking $\mu_c$; the microfibril fraction $\vartheta_f$; and the mean orientation of the microfibrils $a_o$ along with their degree of anisotropy $\xi$ (see 'Supplementary materials and methods' in Appendix 1 for model derivation and parameter definition). The constitutive model was implemented in a 3D FE framework for large deformations, using a meristematic cell with idealised dimensions (*Figure 2A*) whose instantaneous response to inflation under turgor pressure was examined.

We first considered an idealised case in which CMFs were oriented fully randomly (isotropic) at all cell faces (*Figure 2A*, left). To simulate the postulated RAB-A5c-mediated modification of cell wall properties at the edge, we varied the shear moduli $\mu_m$, $\mu_f$, and $\mu_c$ at cell edges from 10-fold less to 10-fold more rigid in comparison to cell faces. As the apparent shear modulus in our current 3D model is a combination of all shear moduli $\mu_m$, $\mu_f$, and $\mu_c$, this approach effectively changed the apparent shear modulus by factor 10, following our previous approach in a 2D linear elastic FE model (*Kirchhelle et al., 2016*). Reduction of shear moduli at edges caused increased displacement at the outer face (i.e., cell swelling; *Figure 2B*, top, D), reproducing the results from a previous simple 2D model (*Kirchhelle et al., 2016*). Increasing shear moduli at cell edges relative to faces decreased displacement at the outer face and was associated with a concentration of stresses at the cell edge domain (*Figure 2—figure supplement 1A*). To simulate our experimental observations regarding CMF reorganisation, we next introduced parallel (anisotropic) CMFs at longitudinal walls (*Figure 2A*, right). Anisotropy at cell faces reduced the displacement at the outer face caused by modification of shear moduli at cell edges (*Figure 2B*, bottom, D). While we did not observe increased CMT anisotropy at the inner faces in wild-type lateral roots, transverse anisotropy at inner faces has been reported for both CMT and CMF arrays in the primary root epidermis (*Panteris et al., 2013*; *Sugimoto et al., 2003*). We therefore tested the effect of changing cellulose anisotropy at the outer face in cells which had transverse anisotropic arrays at the inner anticlinal and periclinal longitudinal face (*Figure 2A*, middle). Introducing anisotropy at just the outer face was sufficient to counter-act cell swelling related to changes in edge properties (*Figure 2B*, middle, D). These findings are consistent with our hypothesis that increased cell wall anisotropy at faces can partially compensate cell swelling induced by local softening of the cell wall at edges.

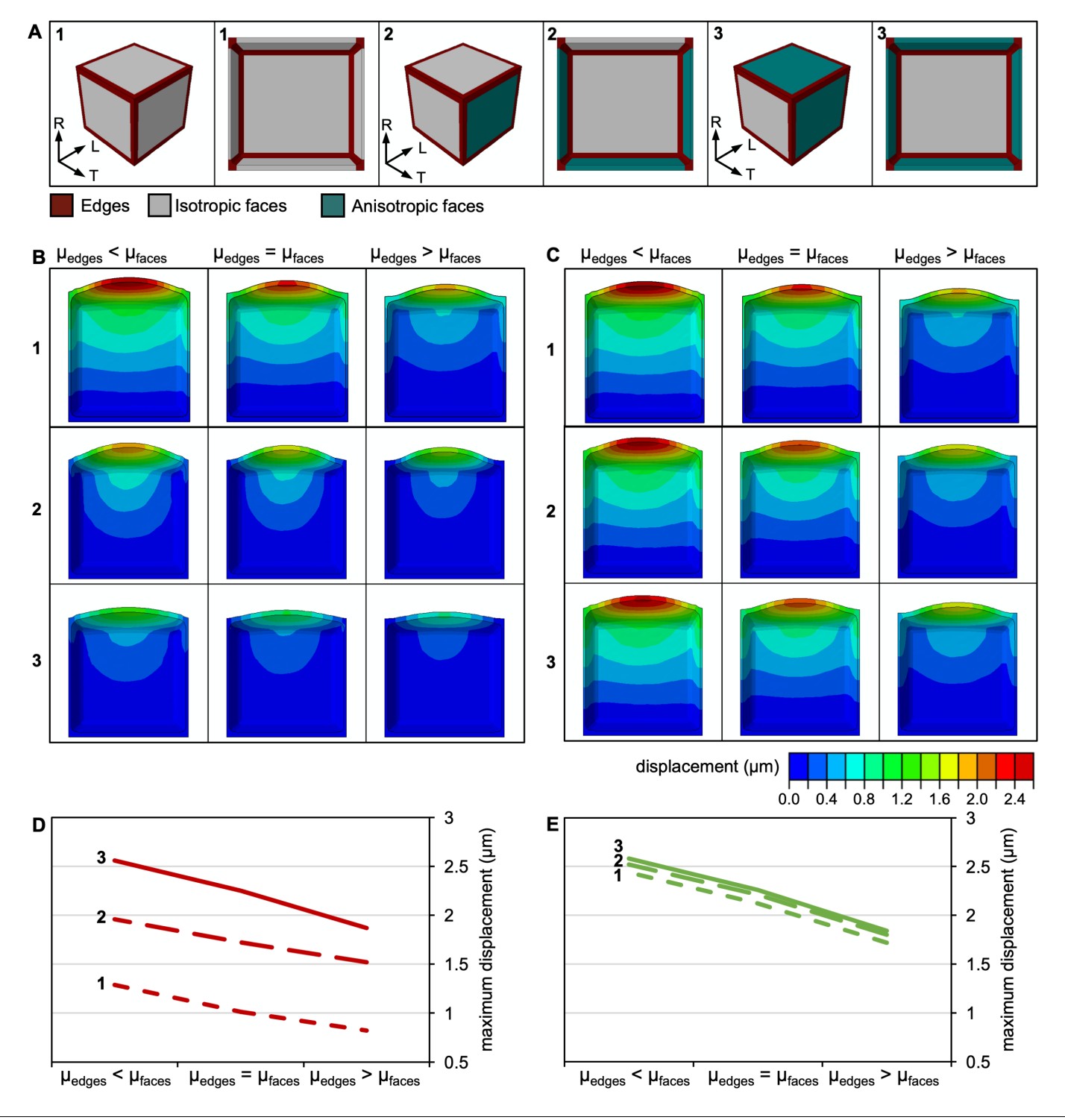

**Figure 2.** A 3D FE model of a meristematic root cell. (A) Morphology of the uninflated model from the outside (left) and in section (right) of a cell with isotropic CMFs at all faces (left, 1), isotropic CMFs at transverse anticlinal and the outer periclinal face, and anisotropic CMFs at longitudinal anticlinal and the inner periclinal face (middle, 2), and isotropic CMFs at transverse anticlinal, and anisotropic CMFs at longitudinal anticlinal and periclinal faces (right, 3). Edges are colour-coded in red, isotropic faces in grey, and anisotropic faces in teal. R: radial, L: longitudinal, T: transverse. (B,C) Effect of selective increase or reduction by factor 10 of shear moduli shear moduli $\mu_m$, $\mu_f$, and $\mu_c$ at cell edges ($\mu_{edges}$) compared to shear moduli at faces ($\mu_{faces}$) on cell morphology for cells in which CMFs were oriented as described in 1,2, and 3 in (A). Cells are colour coded for displacement. (B) idealised cases with FA of 0 (fully isotropic) and 1 (fully anisotropic); (C) cases with FA corresponding to experimentally determined CMT orientation, with FA = 0.08

*Figure 2 continued on next page*

*Figure 2 continued*

corresponding to the isotropic and FA = 0.2 corresponding to the anisotropic case. At faces, $K$ = 10 GPa, $\mu_m$ = 18 MPa, $\mu_f$ = 1.2 GPa, $\mu_c$ = 18 MPa and $\vartheta_f$ = 0.5 in all cases. (D,E) Plots showing maximum displacement of cell models as those shown in (B,C).

DOI: https://doi.org/10.7554/eLife.47988.009

The following figure supplements are available for figure 2:

**Figure supplement 1.** Stress distribution in *in silico* meristematic cells.

DOI: https://doi.org/10.7554/eLife.47988.010

**Figure supplement 2.** Effect of varying model parameters.

DOI: https://doi.org/10.7554/eLife.47988.011

In the idealised cases considering either fully isotropic or fully anisotropic CMFs at cell faces, the effect of CMF orientation on cell swelling outweighed the effects of variations in $\mu_c$ and in shear moduli at edges relative to faces (*Figure 2—figure supplement 2A-C, E*). However, these idealised cases are unlikely to represent the real cell wall accurately, as (1) new CMFs are not deposited perfectly parallel, (2) cell walls contain CMFs laid down during previous stages of development at different orientations, and (3) CMFs in outer layers can change their orientation towards more longitudinal orientations during growth (*Anderson et al., 2010*). Therefore, we used our experimentally determined values for CMT anisotropy in meristematic cells in the presence and absence of RAB-A5c[N125I] (0.2 and 0.08 respectively; *Figure 1C*) as a proxy for CMF anisotropy to ask whether a reduction of cell swelling could still be observed in these more realistic cases. This 2.5-fold increase in anisotropy was sufficient to reduce cell swelling in all cases but one (*Figure 2—figure supplement 2D*), although its contribution towards cell swelling was much smaller relative to variations in $\mu_c$ and in shear moduli at edges relative to faces. We also tested the effect of varying absolute values of the model parameters $\vartheta_f$, $\mu_c$, $\mu_m$, and $\mu_f$, and $K$ (*Figure 2—figure supplement 2*) and found that across all conditions tested, reduction of stiffness at edges enhanced cell swelling, and introduction of anisotropy reduced cell swelling. Taken together, our *in silico* results predict that RAB-A5c acts through locally modifying cell wall properties at edges, and the observed changes in CMT/CMF organisation are secondary effects that can partially compensate for inhibition of RAB-A5c function.

## Simultaneous disruption of RAB-A5c-function and CMT/CMF reorganisation causes a synergistic growth phenotype

To test whether reorganisation of CMTs and CMFs could partially compensate for inhibition of RAB-A5c function as predicted by our model, we investigated how genetic and pharmacological perturbations of either CMT or CMF organisation affected the RAB-A5cNI phenotype. *ectopic root hair 3–3* (*erh3-3*; *Webb et al., 2002*) causes a lesion in the microtubule-severing protein KATANIN p60 (KTN), whose activity at CMT crossover sites is required to establish ordered (anisotropic) CMT arrays (*Burk and Ye, 2002*; *Zhang et al., 2013*). To quantitatively compare the effect of RAB-A5c inhibition on anisotropic growth in the presence and absence of functional KTN, we measured mean lateral root diameter as a proxy for radial cell swelling in *RPS5a > Dex > RAB-A5c[N125I]* and *RPS5a > Dex > RAB-A5c[N125I] erh3-3* lines. Induction of RAB-A5c[N125I] in the wild-type background for 72 hr caused a relative increase of the mean lateral root diameter by 29% (*Figure 3A,B,I*). In the *erh3-3* background, radial swelling was significantly more severe, with a relative increase of mean lateral root diameter by 44% (*Figure 3A,B,I*). This finding indicated that as predicted, CMT reorganisation counteracted cell swelling caused by inhibition of RAB-A5c function.

Loss of KTN function has also been associated with partial reduction in cellulose content (*Burk et al., 2001*), which may contribute to the synergistic phenotype observed. However, in our computational model, reducing cellulose content by half through changing the fibre fraction $\vartheta_f$ had only a minor effect on cell swelling relative to changes in mechanical properties at edges and anisotropy at faces (*Figure 2—figure supplement 2A*). Furthermore, we also investigated the effect of inhibiting CMT reorganisation through pharmacological agents: the microtubule-stabilising agent taxol (*Morejohn and Fosket, 1991*; *Schiff et al., 1979*; *Baskin et al., 1994*), which reduces CMT array dynamics, and the microtubule-depolymerising agent oryzalin (*Hugdahl and Morejohn, 1993*; *Morejohn et al., 1987*), which causes CMT arrays to fragment or fully depolymerise (*Baskin et al., 1994*). Both drugs could prevent the establishment of anisotropic CMT arrays in meristematic lateral

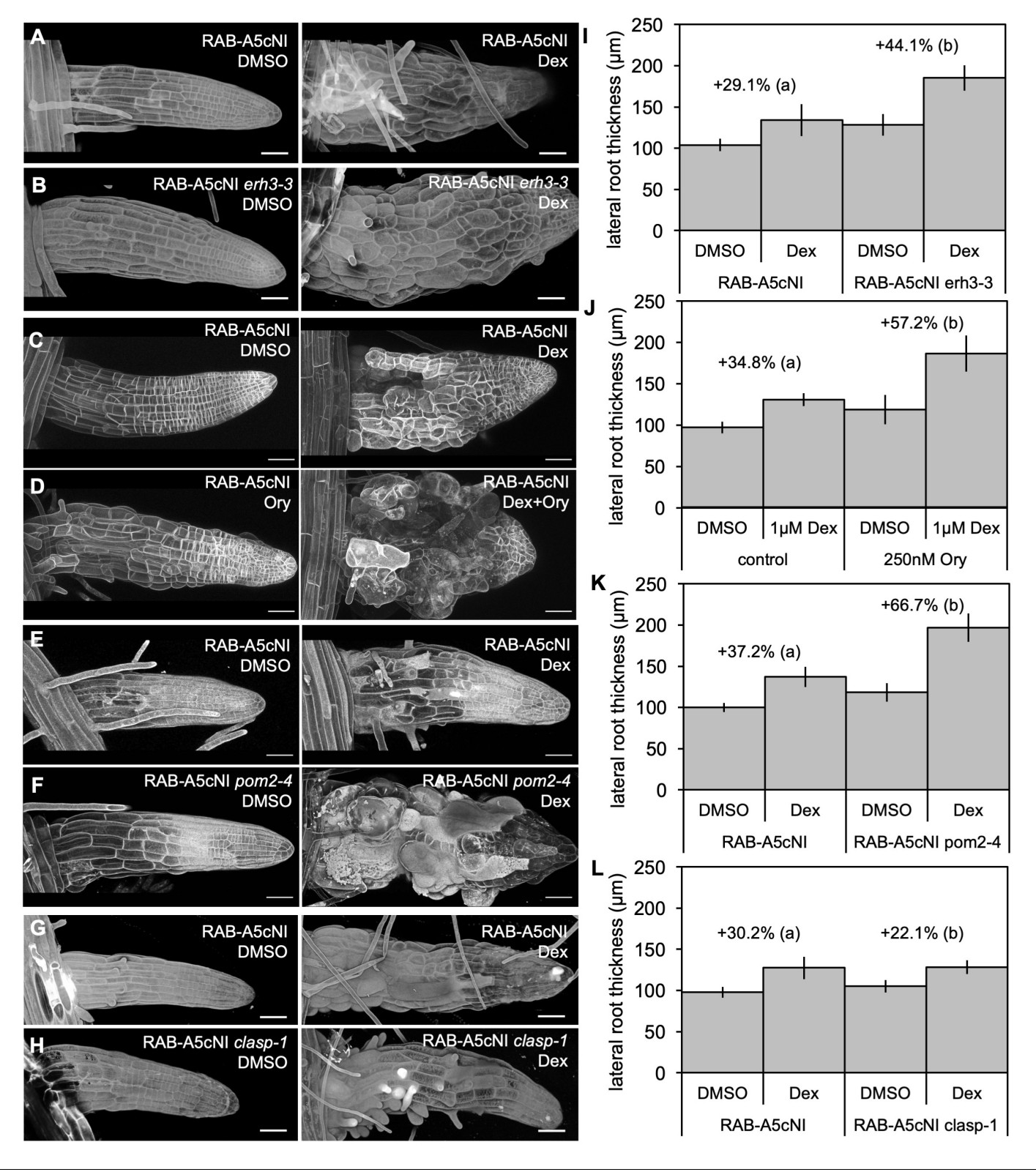

**Figure 3.** Phenotypic interactions between microtubule mutants and RAB-A5c[N125I]. (**A–H**) Confocal stacks of lateral roots expressing *RPS5a > Dex > RAB-A5c[N125I]* in wild-type (**A,C,D,E,G**), *erh3-3* (**B**), *pom2-4* (**F**), or *clasp-1* (**H**) backgrounds; 3 days after transfer to 0.1% DMSO (left) or 1 µM Dex (right), in (**D**), plates additionally contained 250 nM Ory. Cell walls were stained with Propidium Iodide (**A,B,E–H**), or cell outlines were visualised using the plasma membrane marker YFP:NPSN12 (**C,D**). Images are snapshots from MorphoGraphX or maximum intensity projetions of confocal stacks. Scale

*Figure 3 continued on next page*

*Figure 3 continued*

bars 50 μm. (**I–L**) Mean diameter of lateral roots such as those shown in (**A–H**) (n ≥ 21). Difference in diameter (%) between DMSO and 1 μM Dex treatments for each genotype noted above respective columns. Two-way ANOVA and *post hoc* Tukey's test: same letter indicates no significant difference in relative diameter increase (p≥0.05), different letters indicate significant difference (p<0.05).

DOI: https://doi.org/10.7554/eLife.47988.013

The following source data and figure supplements are available for figure 3:

**Source data 1.** Quantification of lateral root thickness in microtubule mutants in the absence or prasence of RAB-A5c[N125I].

DOI: https://doi.org/10.7554/eLife.47988.016

**Figure supplement 1.** Synergistic effect of microtubule inhibitors and RAB-A5c[N125I] expression on lateral root morphology.

DOI: https://doi.org/10.7554/eLife.47988.014

**Figure supplement 1—source data 1.** Quantification of lateral root thickness in response to treatment with oryzalin and taxol in the absence or presence of RAB-A5c[N125I].

DOI: https://doi.org/10.7554/eLife.47988.015

root cells expressing the inhibitory mutant RAB-A5c[N125I] (*Figure 3—figure supplement 1A, F*; note subsaturating levels of oryzalin were used so CMT arrays were partially fragmented rather than fully depolymerised). The increase in mean lateral root diameter in *RPS5a>Dex>RAB-A5c[N125I]* lines after treatment with subsaturating or saturating concentrations of Dex (9% and 35%, *Figure 3— figure supplement 1G, J*) was significantly increased in the presence of both taxol (49% and 51%, respectively, *Figure 3—figure supplement 1H, K*) and oryzalin (52% and 57%, respectively; *Figure 3C, D, J*; *Figure 3—figure supplement 1I, L*). Therefore, preventing reorganisation of CMTs into anisotropic arrays through either genetic or pharmacological perturbations rendered lateral roots hypersensitive to inhibition of RAB-A5c function.

According to our computational model, reorganisation of CMTs counteracts cell swelling in the presence of RAB-A5c[N125I] through the increased anisotropy of newly deposited CMFs. To confirm that the role of CMT arrays in guiding CMF deposition and not an unrelated CMT function was required to counteract cell swelling, we examined the phenotypic interaction between *RPS5a > Dex > RAB-A5c[N125I]* and *pom-pom2-4* (*pom2-4*; *Bringmann et al., 2012*). *pom2-4* is deficient in POM-POM2/CELLULOSE-SYNTHASE INTERACTIVE PROTEIN1 (POM2/CSI1), a protein linking CMTs and cellulose synthase complexes (*Bringmann et al., 2012*; *Li et al., 2012*). In *pom2-4*, CSC trajectories are no longer aligned with CMTs, allowing us to differentiate between CMT reorganisation and orientation of CMF deposition. Similar to *erh3-3*, induction of RAB-A5c[N125I] in the *pom2-4* background caused a significantly larger increase in mean lateral root diameter compared to the wild-type background (67% *vs.* 37%, *Figure 3E,F,K*). This indicated that the effect of CMTs on CMF orientation and not another CMT function was required to compensate for inhibition of RAB-A5c function.

If perturbing the establishment of anisotropic CMT arrays enhanced the RAB-A5c[N15I] phenotype, premature establishment of anisotropic CMT arrays may be sufficient to partially suppress cell swelling. To test this prediction, we investigated the effect of *RPS5a > Dex > RAB-A5c[N125I]* in the *clasp-1* background (*Ambrose et al., 2007*), in which CMT arrays in meristematic root cells display increased anisotropy in transverse orientation (*Ambrose et al., 2011*; *Ambrose et al., 2007*). *clasp-1* lateral roots have a significantly larger diameter than wild-type roots in the absence of RAB-A5c [N125I], but in the presence of RAB-A5c[N125I], mean lateral root diameter was similar in *clasp-1* and wild-type backgrounds. Therefore, the relative increase in lateral root diameter in *clasp-1* comparted to wild-type lateral roots was significantly reduced (22% *vs.* 30%, *Figure 3G,H,L*), indicating that premature alignment of CMTs into anisotropic arrays may indeed partially compensate RAB-A5c[NI]-related cell swelling.

Taken together, these experimental results confirm the predictions of our computational model that the observed increase of CMT and CMF anisotropy in the presence of RAB-A5c[N125I] is a compensatory response partially counteracting the cell swelling caused by inhibition of RAB-A5c function.

## YFP:RAB-A5c is associated with CMTs at cell edges

Edge-localisation of YFP:RAB-A5c compartments was shown to be sensitive to pharmacological depolymerisation of microtubules (*Kirchhelle et al., 2016*). To further investigate the relationship

between RAB-A5c and CMTs, we introduced the microtubule markers p35S::RFP:MBD and pUBQ1:: RFP:TUB6 (*Ambrose et al., 2011*) into YFP:RAB-A5c lines. Confocal stacks of lateral roots expressing either marker combination revealed that YFP:RAB-A5c compartments at cell edges were associated with CMTs (*Figure 4A–C*; *Figure 4—figure supplement 1*; *Figure 4—Videos 1–3*). At edges that were densely labelled with YFP:RAB-A5c, CMTs were always present – perhaps not surprising considering the confined space at a cell edge. However, YFP:RAB-A5c compartments were also sometimes found on cell faces away from the edge, where they were also associated with the ends of microtubules (*Figure 4B*, white arrows). Furthermore, in preprophase cells, where CMTs form the distinctive preprophase band (PPB), YFP:RAB-A5c compartments were progressively restricted to the points on the geometric edges of the cell where the PPB intersected (*Figure 4B,C*, arrowheads; *Figure 4—figure supplement 1*) indicating that YFP:RAB-A5c localisation at cell edges was confined to regions where CMTs were present. After the mitotic spindle was formed, YFP:RAB-A5c was no longer associated with microtubules or cell edges but labelled punctae dispersed throughout the cytoplasm (*Figure 4D*; *Figure 4—figure supplement 1*, *Figure 4—video 3*) before relocating to the cell plate during cytokinesis as previously reported (*Kirchhelle et al., 2016*).

## CMT array structure influences YFP:RAB-A5c pattern

Colocalisation analysis also revealed that not all microtubules at cell edges were associated with YFP:RAB-A5c compartments (*Figure 2B,C*, blue arrows), indicating the presence of CMTs alone may not be sufficient to recruit RAB-A5c to a cell edge. Nevertheless, CMT arrays may influence the relative distribution at different cell edges because in the elongation zone, where CMT arrays are strongly transverse, only longitudinal edges are labelled by YFP:RAB-A5c (*Kirchhelle et al., 2016*). To test this inference, we analysed YFP:RAB-A5c localisation in mutants with defects in CMT array organisation (*Figure 4E–J*, *Figure 4—figure supplement 2*). These mutants included *clasp-1* (*Ambrose et al., 2007*; *Ambrose and Wasteneys, 2008*) and *erh3-3* (*Webb et al., 2002*) described above as well as *spiral3* (*spr3*; *Nakamura and Hashimoto, 2009*), which carries a missense mutation in the γ-tubulin complex protein 2 (GCP2) resulting in a CMT array with a left-handed helical twist (*Nakamura and Hashimoto, 2009*).

3D quantitative analysis revealed that in meristematic cells of wild-type lateral roots, YFP:RAB-A5c was not evenly distributed along all edges of the cell, but was significantly enriched at the longitudinal edges in comparison to transverse edges (*Figure 4E,I*; *Figure 4—figure supplement 2A*). This pattern changed in the *clasp-1, erh3-3,* and *spr3* backgrounds. In *clasp-1*, YFP:RAB-A5c was significantly enriched at the longitudinal edges in comparison to the wild-type. In contrast, both *erh3-3* and *spr3* caused a significant reduction in relative YFP:RAB-A5c intensity at longitudinal edges and a significant increase at transverse edges. As all three mutations also affected cell geometry, we considered the altered YFP:RAB-A5c localisation pattern might be related to such changes. We quantified mean edge length, and found that all mutants had significantly shorter longitudinal and longer transverse walls than the wild-type (*Figure 4J*; *Figure 4—figure supplement 2B*). However, while cell dimensions were statistically indistinguishable between the three mutants, they had opposing effects on YFP:RAB-A5c localisation, indicating that changes in cell geometry were not causal for the observed effects on YFP:RAB-A5c localisation.

Since *clasp-1* causes increased transverse CMT anisotropy in meristematic root cells (resulting in a relative enrichment of microtubules crossing longitudinal edges) and mutations in *erh3-3* delay establishment of transverse CMT anisotropy in older cells, we conclude that CMTs are necessary for RAB-A5c compartment localisation at edges, but on their own, insufficient to explain their distribution at different edges.

## Meristematic lateral root cells exhibit anisotropic growth

Edge localisation of RAB-A5c is most pronounced in meristematic lateral root cells, where CMT arrays are relatively disordered (*Figure 1A,C*). In contrast to unidirectionally growing cells on the cylindrical part of the root, these cells are located on the tapering root apex and therefore have to accommodate a degree of radial and circumferential growth. To investigate 3D growth patterns in these cells, we quantified total volume growth as well as growth in the longitudinal, radial, and circumferential directions in lateral root epidermal cells. To distinguish meristematic cells from the more rapidly elongating transition and elongation zone cells, we used a threshold of 1.5-fold volume

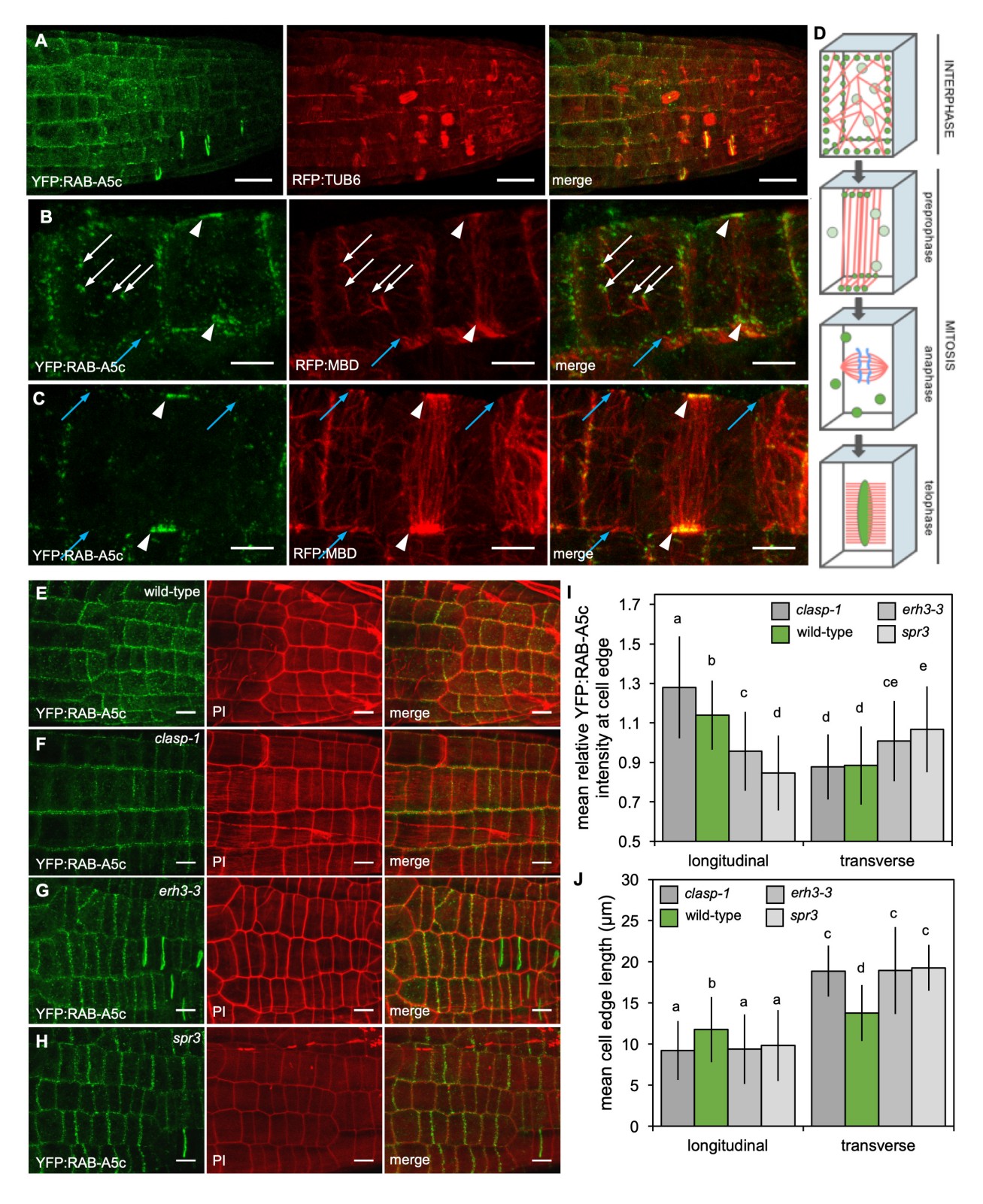

**Figure 4.** YFP:RAB-A5c localisation and CMT organisation. (**A**) Maximum intensity projections of 3D confocal stacks from lateral roots expressing YFP: RAB-A5c (left) and RFP:TUB6 (middle). Scale bar 20 µm. (**B,C**) Maximum intensity projections of 3D confocal stacks from lateral roots expressing YFP: RAB-A5c (left) and RFP:MBD (middle). Scale bars 5 µm. White arrows indicate YFP:RAB-A5c compartments at cell faces co-localising with RFP:MBD, blue arrows indicate CMTs at cell edges not labelled with YFP:RAB-A5c, white arrowheads indicate preprophase bands. (**D**) Schematic summary of RAB-

*Figure 4 continued on next page*

*Figure 4 continued*

A5c localisation in relation to microtubule localisation at different stages of the cell cycle based on images such as those shown in (**A–C**) and S6. (**E–H**) Maximum intensity projections of confocal stacks of lateral roots expressing YFP:RAB-A5c either in wild-type (**E**), *clasp-1* (**F**), *erh3-3* (**G**), and *spr3* (**H**) backgrounds. Cell walls were stained with propidium iodide (PI). Scale bars 10 μm. (**I,J**) Plots showing relative enrichment of YFP:RAB-A5c intensity at an edge normalised against mean edge intensity in the respective cell (**I**) and mean transverse and longitudinal edge length in those cells (**J**) for lateral roots like those shown in (**E–H**). (n ≥ 103 cells for each genotype; two-way ANOVA and *post hoc* Tukey's test: same letter indicates no significant difference (p≥0.05), different letters indicate significant difference (p<0.01).

DOI: https://doi.org/10.7554/eLife.47988.017

The following video, source data, and figure supplements are available for figure 4:

**Figure supplement 1.** YFP:RAB-A5c and RFP:TUB6 dynamics during lateral root development.

DOI: https://doi.org/10.7554/eLife.47988.018

**Figure supplement 2.** YFP:RAB-A5c localisation in microtubule mutants.

DOI: https://doi.org/10.7554/eLife.47988.019

**Figure supplement 2—source data 1.** Quantification of YFP:RAB-A5c intensity at cell edges in microtubule mutant backgrounds.

DOI: https://doi.org/10.7554/eLife.47988.020

**Figure 4—video 1.** 3D maximum intensity projections of 3D confocal stacks from lateral roots expressing YFP:RAB-A5c (top, green) and RFP:MBD (middle, red).

DOI: https://doi.org/10.7554/eLife.47988.021

**Figure 4—video 2.** 3D maximum intensity projections of 3D confocal stacks from lateral roots expressing YFP:RAB-A5c (top, green) and RFP:MBD (middle, red).

DOI: https://doi.org/10.7554/eLife.47988.022

**Figure 4—video 3.** Maximum intensity projections of a 4D confocal stack from a lateral root expressing YFP:RAB-A5c (green) and RFP:TUB6 (red) at consecutive 40 min time intervals.

DOI: https://doi.org/10.7554/eLife.47988.023

growth over 6 hr (*Figure 5—figure supplement 1*). This analysis showed that although total growth rates varied markedly between the two cell populations, in both cases the total volume growth was mostly accounted for by the longitudinal growth vector alone (*Figure 5A,B*). Thus despite their lack of transverse CMT anisotropy, meristematic cells can sustain highly anisotropic longitudinal growth.

## Discussion

In this study, we examined the functional relationship of CMT organisation and RAB-A5c activity at cell edges during lateral root development. A major and surprising finding was that the loss of growth anisotropy caused by RAB-A5c inhibition was associated with increased CMT and CMF anisotropy in meristematic cells of lateral roots. Our data suggest that RAB-A5c contributes to cell growth anisotropy through a mechanism independent of CMT or CMF organisation. Based on our *in silico* data, we propose that this mechanism involves local modification of cell wall stiffness at edges, which can be partially compensated by increased cell wall anisotropy at the faces. Correspondingly we showed experimentally that increased CMT anisotropy could partially compensate the cell swelling phenotype, whereas loss of CMT and CMF reorganisation synergistically enhanced the RAB-A5c [N125I] phenotype. Feedback from cell wall properties to CMT array organisation has been described before: for example, CMT arrays have been shown to react directly to changes in mechanical stresses (*Hamant et al., 2008*; *Uyttewaal et al., 2012*; *Uyttewaal et al., 2010*). It has also been suggested that CMT arrays can be actively altered through changed expression levels of microtubule-interacting proteins when cell wall composition is altered, as in the xyloglucan-deficient *xxt1 xxt2* mutant (*Xiao and Anderson, 2016*). Finally, CMT array organisation is also believed to be influenced by cellulose synthesis (*Fisher and Cyr, 1998*; *Paredez et al., 2008*). While our data indicate RAB-A5c does not act though CMTs, its localisation to cell edges is sensitive to CMT array organisation, and pharmacological depolymerisation of CMTs abolishes edge-localisation (*Kirchhelle et al., 2016*). This indicates CMTs are an integration point for both mechanisms organising anisotropic growth, and also offers a possible explanation for the previously postulated role of CMTs in a mechanism of growth anisotropy that acts independently of CMF orientation (*Sugimoto et al., 2003*): namely, through organising a secretory pathway to the cell edge.

Meristematic cells in lateral roots maintain anisotropic growth despite largely isotropic CMT/ CMFs at the outer periclinal face (*Figure 1A,C–E*, S1 and , S2A). It is possible that parallel CMT/

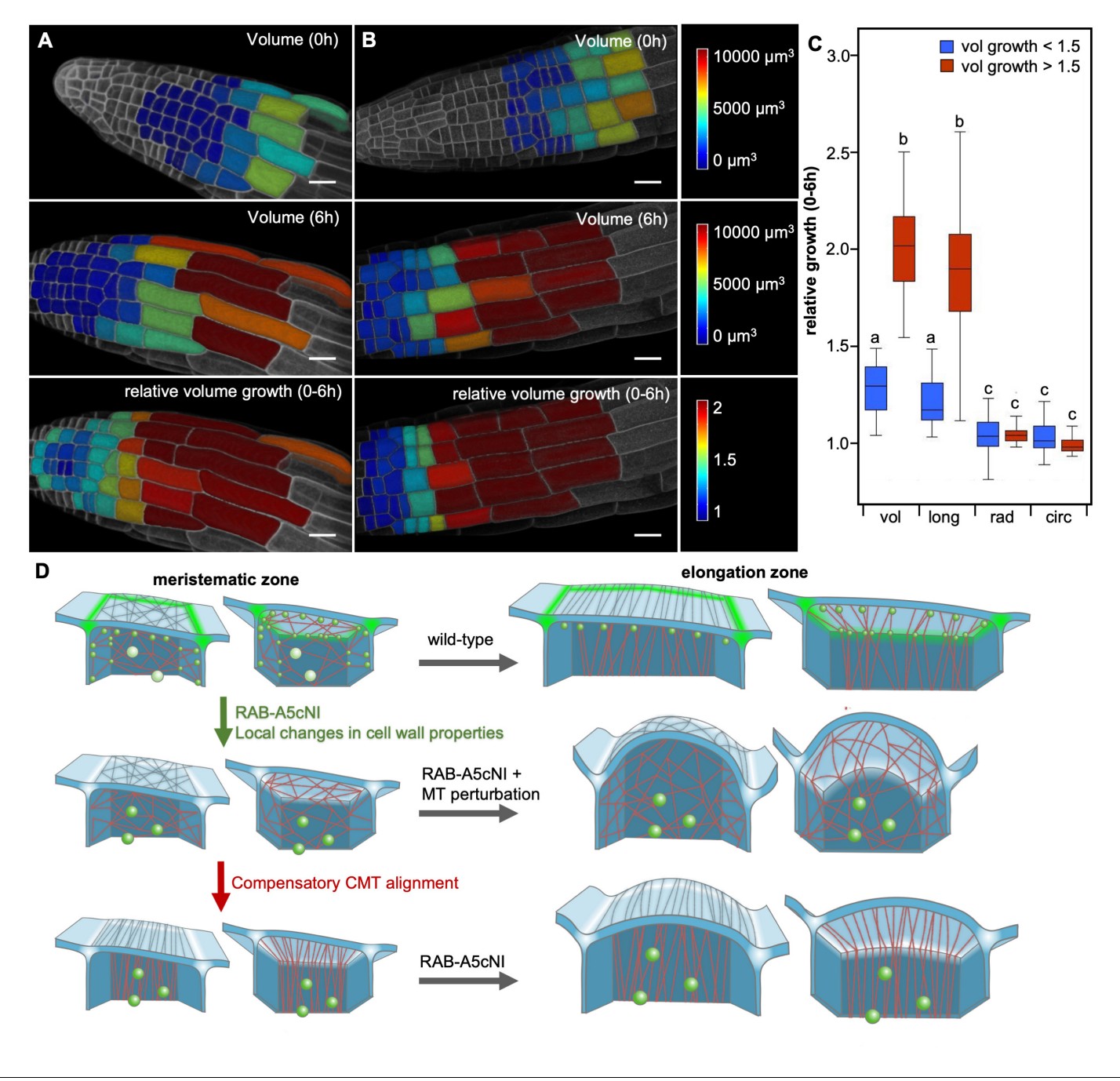

**Figure 5.** Two mechanisms drive growth anisotropy in meristematic lateral root cells. (A,B) Snapshots from time-lapse confocal series of two lateral roots expressing YFP:NPSN12 at 0 hr (top) and 6 hr (middle, bottom). Epidermal cells were segmented in 3D using MorphoGraphX (*Barbier de Reuille et al., 2015*) and are colour-coded for absolute volume (top, middle) or relative volume growth (bottom). (C) boxplot showing relative volume (vol), longitudinal (long), radial (rad), and circumferential (circ) growth for cells shown in (A,B). Cells were split up in two populations based on volume growth to separate slow-growing meristematic cells (growth <1.5, n = 51) from more rapidly growing elongation zone cells (growth >1.5, n = 31). Two-way ANOVA and *post hoc* Tukey's test: same letter indicates no significant difference (p≥0.05), different letters indicate significant difference (p<0.01). Note that while cells grow substantially faster once they have entered the elongation zone, growth is predominantly longitudinal in both meristematic and elongation zone cells, and there is no significant difference in radial or circumferential growth between both cell populations. (D) Proposed model of growth anisotropy regulation in epidermal lateral root cells. Top: in wild-type meristematic cells, CMTs (red) and consequently CMFs (grey) are largely isotropic. Growth anisotropy is predominantly conferred through local modification of cell wall properties at edges (green) through a RAB-A5c-dependent trafficking pathway. In elongation zone cells, CMTs and CMFs are aligned transverse anisotropic, further contributing to anisotropic growth. Middle/Bottom: inhibition of RAB-A5c function through RAB-A5c[N125I] abolishes local modification of cell wall properties at edges, leading to the lack

*Figure 5 continued on next page*

*Figure 5 continued*

of a mechanism promoting anisotropic growth in meristematic cells. This can be partially compensated through premature transverse alignment of CMTs, leading to moderate swelling of cells. If compensatory CMT alignment is prevented through genetic or pharmacological means, cells lack both mechanisms promoting growth anisotropy, and cells swell dramatically.

DOI: https://doi.org/10.7554/eLife.47988.024

The following source data and figure supplements are available for figure 5:

**Source data 1.** Quantification of directional growth in epidermal cells of lateral roots.

DOI: https://doi.org/10.7554/eLife.47988.027

**Figure supplement 1.** Cell growth rates in lateral roots.

DOI: https://doi.org/10.7554/eLife.47988.025

**Figure supplement 2.** YFP:RAB-A5c localisation in young primary roots.

DOI: https://doi.org/10.7554/eLife.47988.026

CMF organisation at inner faces contribute to growth direction control instead, as described for hypocotyl epidermal cells (*Crowell et al., 2011*). Parallel CMF arrays at the inner periclinal face have been described in primary roots (*Sugimoto et al., 2000*; *Panteris et al., 2013*). Our findings presented here indicate that in lateral roots, CMT arrays at the L1/L2 interface are oriented in transverse orientation more frequently that those at the outer face, however CMT anisotropy is relatively low. Here, we describe an alternative CMF/CMT-independent mechanism mediated by RAB-A5c which can also control growth direction. We propose the following model (*Figure 5C*): in wild-type meristematic cells, CMTs and consequently CMFs are predominantly isotropic, and growth anisotropy is conferred through local modification of cell wall properties at edges through a RAB-A5c-dependent trafficking pathway. As cells begin to enter the elongation zone, CMTs and CMFs are aligned in transverse direction, further contributing to rapid anisotropic growth. Inhibition of RAB-A5c function through RAB-A5c[N125I] interferes with local modification of cell wall properties at edges, causing loss of growth direction control. This can be partially compensated through premature transverse alignment of CMTs and CMFs, causing cells to only moderately swell. If compensatory CMT or CMF reorganisation is prevented through genetic or pharmacological means, cells lack both mechanisms promoting growth anisotropy, so growth becomes isotropic and cells swell dramatically. This model implies cells use different mechanisms to promote anisotropic growth at different developmental stages: anisotropic growth control in meristematic cells is dominated by RAB-A5c activity at cell edges, whereas cells in the elongation zone largely rely on CMT and CMF anisotropy at faces to control growth anisotropy. This may explain our previous observation that loss of growth anisotropy is most pronounced if RAB-A5c is inhibited in meristematic cells that enter the elongation zone subsequently, whereas cells that were in the transition zone (where parallel CMT arrays are established at the outer face) when RAB-A5c was inhibited maintained anisotropic growth (*Kirchhelle et al., 2016*).

Why might different mechanisms be involved in directional growth control in these different cell populations? This remains an area of speculation but we offer some considerations below. It may be that the simple transverse CMT/CMF mechanism that is adequate to sustain simple rapid cylindrical extension of cells (and roots) in the elongation zone is inadequate for the meristem. The root meristem has a tapering apex that must be organised into a regular cylinder by the time cells enter the transition zone. This may require more flexibility in growth direction and thus in CMT and CMF arrangement. Secondly, the meristem must contend with multiplicative and formative cell divisions, each of which will have imperfections in positioning and insertion angle that must be accommodated to achieve a regular organ geometry. It has been noted previously how variability in growth properties is greater at the cellular scale than the organ scale and is possibly required for the high degree of uniformity observed at the higher scale (*Hong et al., 2016*; *Meyer and Roeder, 2014*). However, while there may be an increased requirement for flexibility in the growth vector in meristematic lateral root cells, they still grow largely anisotropically even in the absence of transverse CMF arrays (*Figure 5A,B*).

One possible explanation for why the CMT/CMF mechanism may be inadequate to provide sufficient anisotropy while maintaining flexibility arises from the observation that cell wall thickness in the root remains relatively constant as cells progress through the meristem (*Dyson et al., 2014*). As changes in facial anisotropy through CMF orientation rely on deposition of new material, it follows they are largely growth-rate dependent, and may therefore be inefficient in slow-growing

meristematic cells. It is conceivable that RAB-A5c acts through a mechanism that is not growth-rate dependent, for example through the secretion of cell wall modifying enzymes that can locally change the properties of the cell wall (*Peaucelle et al., 2015*). Furthermore, our 3D model predicts that cells with largely isotropic CMF arrays will be particularly sensitive to alterations in mechanical properties at cell edges (*Figure 2C*), whereas CMF anisotropy will dominate in cells with highly aligned CMF arrays (*Figure 2B*). Therefore, it is possible that premature establishment of highly anisotropic arrays will render meristematic cells devoid of the required flexibility in regulating growth directionality, whereas a RAB-A5c-mediated mechanism acting at cell edges may provide the requisite anisotropic growth control while leaving CMT and cell wall anisotropy sufficiently flexible for regular organ morphogenesis.

YFP:RAB-A5c was poorly expressed and generally did not localise to cell edges in older lateral or primary roots (*Kirchhelle et al., 2016*). Why are such roots less dependent on the RAB-A5c-based mechanism? A CMF/CMT-based mechanism might be sufficient to control directional growth in longer roots which do not taper as dramatically as young lateral roots. Alternatively, anisotropic growth in older roots may be aided by the overlying lateral root cap which could mechanically resist radial and circumferential (transverse) growth, favouring longitudinal growth. Indeed, in young primary roots YFP:RAB-A5c was present at cell edges in meristematic cells that were not covered by the lateral root cap but not in adjacent cells lying beneath it (*Figure 5—figure supplement 2*). Furthermore, genetic ablation of the root cap results in cellular growth defects in the primary root epidermis that are reminiscent of loss of RAB-A5c activity (*Tsugeki and Fedoroff, 1999*). It may be significant that young lateral root primordia that express RAB-A5 lack a well-developed overlying root cap to counteract radial swelling of meristematic cells. These findings, together with our previous observation that RAB-A5c is predominantly expressed in the lateral root epidermis (*Kirchhelle et al., 2016*), suggest a requirement for RAB-A5c specifically in cells at the organ surface.

# Materials and methods

**Key resources table**

| Reagent type (species) or resource | Designation | Source or reference | Identifiers | Additional information |
|---|---|---|---|---|
| Gene (*Arabidopsis thaliana*) | RAB-A5c/ARA4 | PMID: 1748311 PMID: 26906735 | AT2G43130 | |
| Gene (*Arabidopsis thaliana*) | KTN | PMID: 11283338 | AT1G80350 | |
| Gene (*Arabidopsis thaliana*) | CLASP | PMID: 17873093 | AT2G20190 | |
| Gene (*Arabidopsis thaliana*) | CSI1/POM2 | PMID: 20616083 | AT2G22125 | |
| Gene (*Arabidopsis thaliana*) | TUB6 | PMID: 1498609 | AT5G12250 | |
| Gene (*Arabidopsis thaliana*) | GCP2 | PMID: 17714428 | AT5G05620 | |
| Gene (*Arabidopsis thaliana*) | CESA1/RSW1 | PMID: 9445479 | AT4G32410 | |
| Strain, strain background (*Arabidopsis thaliana*) | WT; Wild- Type; Col0 | NASC | Nasc stock number: N1093 | |
| Genetic reagent (*Arabidopsis thaliana*) | *erh3-3* | PMID: 11782406 | | |
| Genetic reagent (*Arabidopsis thaliana*) | *spr3* | PMID: 19509058 | | |
| Genetic reagent (*Arabidopsis thaliana*) | *clasp-1* | PMID: 17873093 | | |

*Continued on next page*

*Continued*

| Reagent type (species) or resource | Designation | Source or reference | Identifiers | Additional information |
|---|---|---|---|---|
| Genetic reagent (*Arabidopsis thaliana*) | *pom2-4* | PMID: 22294619 | | |
| Genetic reagent (*Arabidopsis thaliana*) | RPS5a > Dex > RAB-A5c[N125I] | PMID: 26906735 | | |
| Genetic reagent (*Arabidopsis thaliana*) | RAB-A5c::YFP:RAB-A5c | PMID: 26906735 | | |
| Genetic reagent (*Arabidopsis thaliana*) | UBQ10:: YFP:NPSN12 | PMID: 19309456 | | |
| Genetic reagent (*Arabidopsis thaliana*) | UBQ10:: mCherry: NPSN12 | PMID: 19309456 | | |
| Genetic reagent (*Arabidopsis thaliana*) | CLASP::GFP:CLASP | PMID: 17873093 | | |
| Genetic reagent (*Arabidopsis thaliana*) | UBQ1::RFP:TUB:6 | PMID: 21847104 | | |
| Genetic reagent (*Arabidopsis thaliana*) | GCP2::GCP2:3xGFP | PMID: 20935636 | | |
| Genetic reagent (*Arabidopsis thaliana*) | p35S::RFP:MBD | This paper | | |
| Chemical compound, drug | Oryzalin | Sigma-Aldrich | CAS: 19044-88-3 | |
| Chemical compound, drug | Taxol | Sigma-Aldrich | CAS: 33069-62-4 | |
| Chemical compound, drug | Dexamethasone | Sigma-Aldrich | CAS: 50-02-2 | |
| Software, algorithm | Fiji (Fiji is just ImageJ) | PMID: 22743772 | https://imagej.net/Fiji | |
| Software, algorithm | MorphoGraphX | PMID: 25946108 | https://www.mpipz. mpg.de/MorphoGraphX | |

## Plant material and growth conditions

The Columbia ecotype was used throughout. Seeds expressing YFP:RAB-A5c under its native promoter and lines expressing RAB-A5c[N125I] under the control of the Dex-inducible pOp/LhGR expression system (*RPS5a > Dex > RAB-A5c[N125I]*) were described before (*Kirchhelle et al., 2016*). Seeds expressing plasma membrane markers YFP:NPSN12 or mCherry:NPSN12 under the UBIQUITIN10 promoter are part of the WAVE line collection (*Geldner et al., 2009*). Lines expressing GFP: CLASP under its native promoter in a clasp-1 background (*Ambrose et al., 2007*), seeds expressing RFP:TUB6 under a UBIQUITIN1 promoter (*Ambrose et al., 2011*), seeds expressing GCP2:3xGFP under its native promotor (*Nakamura et al., 2010*), and seeds expressing mCherry:CESA1 under its native promoter (*Vain et al., 2014*) have been described before. Mutants clasp-1 (*Ambrose et al., 2007*), erh3-3 (*Webb et al., 2002*), pom2-4 (*Bringmann et al., 2012*) and spr3 (*Nakamura and Hashimoto, 2009*) have been described before. To generate p35S::RFP:MBD expressing plants, Col-0 plants were transformed with p35S::RFP:MBD in gateway vector pK7WGR2 (*Van Damme et al., 2004*) *via* Agrobacterium-mediated transformation (*Clough and Bent, 1998*).

Lateral roots were imaged from seedlings after 7–14 days in a growth chamber (20°C, 16 hr light/ 8 hr dark) on vertically oriented 0.8% Bacto agar (BD Biosciences) plates with half-strength Murashige and Skoog medium (MS, Sigma-Aldrich), and 1% w/v sucrose (pH 5.7) (½ MS). For Dexamethasone (Dex, Sigma-Aldrich) induction and pharmacological treatments, seedlings were germinated and grown for 6–11 days to allow lateral root development and then transferred onto medium containing either the respective drug or an equivalent volume of solvent for up to three days. Dex was applied at 300 nM, 1 µM or 20 µM diluted from a 20 mM stock in DMSO, Oryzalin (Ory, Sigma-Aldrich) was applied at 250 nM diluted from a 10 mM Stock in DMSO and Taxol (Tax, Sigma-Aldrich) was applied at 10 µM diluted from a 10 mM Stock in DMSO.

## Microscopy

All confocal images were acquired using a HCX PL APO CS 20×/0.7 IMM UV lens or a HCX PL APO 63×/1.2 NA lens on a Leica TCS SP5, as described previously (*Kirchhelle et al., 2016*). Time-lapse imaging of lateral roots was performed in imaging chambers as described before (*Kirchhelle and Moore, 2017*).

## Quantitative image analysis

Quantification of YFP:RAB-A5c was performed using MorphographX (*Barbier de Reuille et al., 2015*) in essence as described previously (*Kirchhelle et al., 2016*), with two modifications: (1) the analysis was limited to the region 0–2 µm from the outer surface of each cell, with the region 0–1 µm from the anticlinal wall considered as edge, and (2) longitudinal and transverse walls were considered separately.

To quantify colocalisation between GCP2:GFP and YFP:RAB-A5c, cytosolic background signal was removed from confocal stacks using a hysteresis filter (thresholds based on mean and maximum intensity + 2SD) in Fiji (*Schindelin et al., 2012*). On maximum intensity projections of filtered stacks, longitudinal and transverse edges were manually identified and Mander's colocalisation coefficients (*Manders et al., 1993*) were determined for a region extending 1 µm in each direction from the cell edge using JACoP (Just Another Colocalisation Plugin) in Fiji (*Bolte and Cordelières, 2006*). Cell edges where either fluorophore occupied less than 10% of the pixels were excluded to avoid bias.

Microtubule array anisotropy was quantified either as described before using the FibrilTool plugin in Fiji (*Boudaoud et al., 2014*), or in MorphoGraphX (*Barbier de Reuille et al., 2015*). For MorphographX-based CMT array quantification, 2.5D segmentation was performed as described before (*Barbier de Reuille et al., 2015*; *Kirchhelle et al., 2016*). After segmentation was completed, the RFP:TUB6 stack was imported into MorphoGraphX, filtered with a Gaussian Blur filter with a radius of 0.1 µm, and projected onto the surface (0–1.5 µm from the organ surface). CMT array anisotropy was determined using the Fibril Orientation tool. Maximum ($A_{max}$) and minimum ($A_{min}$) anisotropy values for each interphase meristematic cell were exported from MorphographX as csv files. Dividing cells were excluded from the analysis. Anisotropy values were normalised as follows for each cell: $\frac{(A_{max} - A_{min})}{(A_{max} + A_{min})}$; that is purely isotropic arrays had an anisotropy of 0 and purely anisotropic arrays have an anisotropy of 1.

To assess microtubule orientation at the L1/L2 interface, 3D segmentation was performed as described before (*Barbier de Reuille et al., 2015*; *Kirchhelle et al., 2016*). After segmentation was completed, the RFP:TUB6 stack was imported into MorphoGraphX, filtered with a Gaussian Blur filter with a radius of 0.1 µm, and projected onto the surface (0–1.5 µm from the organ surface). Note that the cell wall thickness at the L1/L2 interface of less than 200 nm is below the z-resolution limit of CLSM, we therefore cannot exclude a contribution of microtubules from the outer face of cortex cells to our measurements at the L1/L2 interface, and therefore use the term 'L1/L2 interface' instead of inner periclinal face.

To quantify root thickness, bright-field images of lateral roots were imported into Fiji, both sides of the root were traced manually along their longitudinal axis, and XY Cartesian coordinates for each pixel on the outline trace were exported as csv files and imported into RStudio (https://www.rstudio.com/). For each pixel on one side, the closest neighbour on the other side was determined and the Euclidian distance between pixels calculated using the nn2 function in the RANN package (https://CRAN.R-project.org/package=RANN). The mean diameter of each root was calculated as the average of all evaluated pixels excluding the tip-most 100 µm of each root to exclude the tapering tip.

## Statistical data analysis and plotting

Two-way ANOVA (analysis of variance) was performed in R using the aov function from the stats package (https://stat.ethz.ch/R-manual/R-devel/library/stats/), Tukey's test was performed in R using the TukeyHSD function from the stats package, Student's t-test and, for samples with unequal variance, Welch's t-test were performed in R using the t.test function from the stats package. Plots were generated in R using the plot function from the graphics package (R Core Team, 2012; https://stat.ethz.ch/R-manual/R-devel/library/graphics/), in Gnumeric (http://www.gnumeric.org/), or in Microsoft Office Excel 2016 (https://products.office.com/en-gb/excel).

## Acknowledgements

The work on this manuscript has been overshadowed by the untimely death of Ian Moore in August 2018. In sorrow, we dedicate this work to his memory. We thank Geoff Wasteneys for providing the CLASP:GFP and RFP:TUB6 lines, Takashi Hashimoto for the *spr3* and GCP2:GFP lines, and Jane Langdale for critical reading of the manuscript. This work was supported by Leverhulme Trust grant RPG-2014–276 and a John Fell Award to IM, BBSRC grant BB/P01979X/1 to IM, CK, and AJ, and a BBSRC doctoral training award, Clarendon Scholarship, and Leverhulme Early Career Fellowship ECF-2017–483 to CK. DG-G and AJ acknowledge funding from the European Union's Seventh Framework Programme (FP7 2007–2013) ERC Grant Agreement No. 306587.

## Additional information

### Funding

| Funder | Grant reference number | Author |
|---|---|---|
| Biotechnology and Biological Sciences Research Council | BB/P01979X/1 | Charlotte Kirchhelle Antoine Jérusalem Ian Moore |
| Leverhulme Trust | RPG-2014-276 | Niloufer G Irani Ian Moore |
| Leverhulme Trust | ECF-2017-483 | Charlotte Kirchhelle |
| John Fell Fund, University of Oxford | | Charlotte Kirchhelle Antoine Jérusalem Ian Moore |
| Seventh Framework Programme | ERC Grant Agreement No. 306587 | Daniel Garcia-Gonzalez Antoine Jérusalem |

The funders had no role in study design, data collection and interpretation, or the decision to submit the work for publication.

### Author contributions

Charlotte Kirchhelle, Conceptualization, Data curation, Formal analysis, Funding acquisition, Validation, Investigation, Visualization, Methodology, Writing—original draft, Project administration, Writing—review and editing; Daniel Garcia-Gonzalez, Software, Investigation, Visualization, Methodology, Writing—review and editing; Niloufer G Irani, Resources, Writing—review and editing; Antoine Jérusalem, Software, Supervision, Funding acquisition, Methodology, Writing—review and editing; Ian Moore, Supervision, Funding acquisition, Investigation, Project administration, Writing—review and editing

### Author ORCIDs

Charlotte Kirchhelle https://orcid.org/0000-0001-8448-6906
Daniel Garcia-Gonzalez http://orcid.org/0000-0003-4692-3508

### Decision letter and Author response

Decision letter https://doi.org/10.7554/eLife.47988.032
Author response https://doi.org/10.7554/eLife.47988.033

## Additional files

### Supplementary files

• Source code 1. Computational model, abaqus subroutine.
DOI: https://doi.org/10.7554/eLife.47988.028

• Transparent reporting form
DOI: https://doi.org/10.7554/eLife.47988.029

## Data availability

Source data files have been provided for Figure 1, Figure 4, Figure 5, Figure 3, Figure 1—figure supplement 2, Figure 1—figure supplement 3, and Figure 3—figure supplement 1. The source code file has been provided for the computational model (Figure 2 and Figure 2—figure supplements 1 and 2).

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

## Appendix 1

DOI: https://doi.org/10.7554/eLife.47988.030

## Supplementary materials and methods

### Computational Model

To adequately describe the mechanical behaviour of the different components of the plant cell wall under turgor pressure (positive on the wall, by convention), both bulk and shear responses need to be considered. Here, we decompose the stress response into volumetric and deviatoric (isochoric) components as:

$$\sigma = \sigma_{\mathrm{vol}} + \sigma_{\mathrm{iso}} \tag{1}$$

where $\sigma$ is the Cauchy stress tensor, and $\sigma_{\mathrm{vol}}$ and $\sigma_{\mathrm{iso}}$ are the volumetric and deviatoric components of the Cauchy stress tensor, respectively. The deformation gradient is similarly multiplicatively decomposed into volumetric and deviatoric components:

$$\mathbf{F} = \mathrm{J}^{\frac{1}{3}}\mathbf{F}^* \tag{2}$$

where $\mathrm{J} = \det(\mathbf{F})$ is the Jacobian and $\mathbf{F}^*$ is the distortional part of the deformation gradient. The volumetric stress response is defined by a linear approximation as:

$$\sigma_{\mathrm{vol}} = \mathrm{K}(\mathrm{J} - 1)\mathbf{I} \tag{3}$$

where $\mathrm{K}$ is the bulk modulus of the material and $\mathbf{I}$ is the second order identity tensor.

The plant cell wall is a composite material dominated by the polysaccharides cellulose, hemicellulose, and pectin, with cellulose microfibrils being considered the main component conferring anisotropy (see Introduction). To test our hypothesis that localised modifications of cell wall properties at cell edges as well as anisotropy at cell faces can significantly contribute to cell geometry, we employed a general energy-based formulation considering the overall response of the cell wall as the combination of an isotropic matrix contribution $\Psi_{\mathrm{m}}$ (dominated by pectins) and an anisotropic fibre contribution $\Psi_{\mathrm{f}}$ (dominated by cellulose microfibrils):

$$\Psi = \Psi_{\mathrm{m}} + \Psi_{\mathrm{f}} \tag{4}$$

The contribution of each phase of the model is weighted by the introduction of the fibre fraction $\vartheta_{\mathrm{f}}$ in the definition of both matrix and fibre energy functions, allowing for the prediction of the mechanical behaviour of cell wall depending on its specific composition.

We consider the cell wall matrix to act as an isotropic hyperelastic material, and choose a Neo-Hookean energy function as a first approximation (*Rivlin, 1948*):

$$\Psi_{\mathrm{m}} = (1 - \vartheta_{\mathrm{f}})\frac{\mu_{\mathrm{m}}}{2}\left(\mathrm{I}_1^* - 3\right) \tag{5}$$

where $\mu_{\mathrm{m}}$ is the shear modulus of the cell wall matrix and $\mathrm{I}_1^* = \operatorname{tr}\mathbf{C}^*$ is the distortional first strain invariant. The distortional right Cauchy-Green deformation tensor is defined by $\mathbf{C}^* = (\mathbf{F}^*)^{\mathrm{T}}\mathbf{F}^*$. Following the usual conventions (see for example *Garcia-Gonzalez et al., 2018*), the constitutive equation that defines the stress response of the matrix reads:

$$\sigma_{\mathrm{m}} = 2\mathrm{J}\mathbf{F}\frac{\partial \Psi_{\mathrm{m}}}{\partial \mathbf{C}}\mathbf{F}^{\mathbf{T}} \tag{6}$$

The fibre contribution is defined following the previous approach of the authors for the axonal contribution in white matter mechanical behaviour (*Garcia-Gonzalez et al., 2018*). Here, the orientation of the fibres and their distribution are introduced through a symmetric structure tensor $\mathbf{A}_{\mathrm{o}}$. This structure tensor depends on a dispersion parameter $\xi$ and the fibre orientation $\mathbf{a}_{\mathrm{o}}$ as:

$$\mathbf{A_o} = \xi\mathbf{I} + (1 - 3\xi)\mathbf{a_o} \otimes \mathbf{a_o} \tag{7}$$

The value adopted by the dispersion parameter $\xi$ determines the degree of anisotropy. This coefficient adopts a value of 0 if the fibres are perfectly aligned, leading thus to a pure anisotropic response, and a value of 1/3 if there is an isotropic distribution of the fibres resulting in a pure isotropic response. The parameter $\xi$ for a specific fibre fractional anisotropy (FA) can be obtained by interpolation through $\xi = -\frac{1}{3}\text{FA} + \frac{1}{3}$. Note that an absence of fibres also leads to $\xi = 1/3$. While we acknowledge this limitation, we assume *Equation 7* as a first approximation of anisotropic distribution in the presence of fibres, as pertains in the plant cell wall.

The average distortional stretch of the fibres $\bar{\lambda}_F^*$ can be defined as:

$$\bar{\lambda}_F^* = \sqrt{\mathrm{I}_{4F}^*} \tag{8}$$

where $\mathrm{I}_{4F}^*$ is the fourth distortional strain invariant that depends on $\mathbf{A_o}$ and $\mathbf{C}^*$ through:

$$\mathrm{I}_{4F}^* = \mathrm{tr}(\mathbf{A_o}\mathbf{C}^*) \tag{9}$$

The free energy function of the fibre contribution is defined by the *standard reinforcing model* (***Nguyen et al., 2007***; ***Wysocki et al., 2010***):

$$\Psi_f = \vartheta_f\left[\frac{\mu_c}{2}\left(\mathrm{I}_1^* - 3\right) + \frac{\mu_f}{4}\left(\mathrm{I}_{4F}^* - 1\right)^2\right] \tag{10}$$

where $\mu_f$ is the shear modulus of the fibre phase. We also introduce an additional shear modulus $\mu_c$, which accounts for cross-linking of cellulose microfibrils either through direct interaction or mediated through hemicellulose. Note that even in highly anisotropic cell walls, such crosslinking has been described (***Cosgrove, 2014***), and omission of this parameter would lead to a complete absence of resistance to deformation perpendicular to fibre direction in a perfectly anisotropic array. As cross-linking sites (or mechanical hotspots) occur in both anisotropic and isotropic microfibril arrays and are substantially shorter than microfibrils themselves (***Cosgrove, 2014***), we consider its contribution to be isotropic. The constitutive equation that defines the stress response of the fibres then reads:

$$\sigma_f = 2\mathrm{J}\mathbf{F}\frac{\partial\Psi_f}{\partial\mathbf{C}}\mathbf{F^T} \tag{11}$$

The constitutive model was implemented in a material subroutine for Abaqus/Implicit (SIMULIA, see http://abaqus.software.polimi.it/v6.14/index.html for documentation) and assigned to the corresponding parts of an idealised cell model. Cells were modelled as cubes measuring 12 µm in radial, circumferential, and longitudinal directions. Anticlinal walls were 0.25 µm thick, periclinal walls 0.5 µm thick. A fillet with a radius of 0.5 µm was added at all edges, the regions of the cell wall adjacent to the fillets were defined as edges (*Figure 2A*, red), and the regions between edges were defined as faces (*Figure 2A*, grey, teal). The cell was meshed with 29,985 linear tetrahedral elements (C3D4).

The model requires the identification of seven parameters: the bulk modulus $\mathrm{K}$; the shear moduli of the cell wall matrix $\mu_m$, cellulose microfibrils $\mu_f$, and crosslinking $\mu_c$; the microfibril fraction $\vartheta_f$; and the mean orientation of the microfibrils $\mathbf{a_o}$ and their degree of anisotropy related to $\xi$. Experimental determination of cell wall mechanical properties is notoriously difficult, and depending on the experimental technique employed, results vary over several orders of magnitude (***Cosgrove, 2016***). Considering there are no reliable data available for meristematic root cells, we adopted recently published values for *Arabidopsis* guard cells of $\mu_m$ = 18MPa, $\mu_f$ = 1.2GPa, and $\mathrm{K}$ = 10GPa (***Woolfenden et al., 2017***). We tested the effect of increasing and decreasing $\mu_m$, $\mu_f$, and $\mathrm{K}$ by factor 2 to assess the the sensitivity of the model parameters (*Figure 2—figure supplement 2*). We found that changes in all three parameters resulted in small changes in maximum displacement of the outer face. We found that changes in $\mu_m$ had a larger effect in isotropic than in anisotropic cases, whereas changes in $\mu_f$ predominantly affected anisotropic cases. This observation from the model is consistent with

the expectation that the matrix may dominate the material response when microfibrils are randomly oriented, whereas microfibrils may dominate in cases where they are ordered. As there were no data available for the shear modulus $\mu_c$, we tested the effect of varying this parameter (*Figure 2—figure supplement 2*) on the overall output of our model. Cell swelling was progressively reduced when $\mu_c$ was increased, but introduction of anisotropy at faces further reduced cell swelling. In the primary cell wall, estimates for cell wall composition vary somewhat, from 14% cellulose + 24% hemicellulose (*Zablackis et al., 1995*) to approximately 50% for the cellulose-hemicellulose fraction of total cell wall mass (*Carpita and Gibeaut, 1993*). Considering that hemicellulose contributes to the fibre fraction in our model through crosslinking, but cellulose is the main load-bearing fibre, we tested our model for $\vartheta_f = 0.25$ and 0.5 to examine whether relative fibre content significantly influenced our findings (*Figure 2—figure supplement 2*). Both tested values yielded very similar results, and we performed subsequent simulations with $\vartheta_f = 0.5$. $\xi$ can adopt values between 0 (perfectly parallel, anisotropic fibre alignment) and 1/3 (fully isotropic fibre alignment). Therefore, $\xi = 0$ corresponds to a FA of 1 following the convention laid out above, and $\xi = 1/3$ corresponds to FA = 0. In addition to these idealised cases, we also tested conditions in which the fibre dispersion parameter $\xi$ corresponded to the experimentally determined CMT anisotropy in meristematic cells in the wild-type background (largely isotropic orientation; FA = 0.08, $\xi = 0.31$) and the *RPS5a>Dex>RAB-A5c[N125I]* background (increased anisotropy, FA = 0.2; $\xi = 0.27$).

