## [Decision Letter]

Thank you for submitting your article "Two mechanisms for regulating directional growth of cells in Arabidopsis lateral roots" for consideration by *eLife*. Your article has been reviewed by three peer reviewers, and the evaluation has been overseen by a Reviewing Editor and Christian Hardtke as the Senior Editor. The reviewers have opted to remain anonymous.

The reviewers have discussed the reviews with one another and the Reviewing Editor has drafted this decision to help you prepare a revised submission.

The conclusion was that your work nicely illustrates that RAB-A5c function at cell edges is associated with cortical microtubules, thereby controlling cellular growth anisotropy. The reviewers were very positive about your work. They discussed among each other and requested the following essential improvements before the paper can be accepted:

Please provide information on what is happening on the inner walls (epidermis/cortex interface), as several studies have indicated that both faces can behave very differently. In addition, we request improvement of the CMT imaging.

Please, have also a look at the detailed reviewer comments below, which may guide you to further improve your manuscript.

*Reviewer #1:*

The manuscript of Kirchhelle et al. provides further insights into the function of RAB-A5C, a member of the largely extended family of RAB11 homologs in plants. The authors had previously reported that this RAB localises in a very specific fashion to particular membrane compartments, localising to cellular edges and that interference with RAB-A5c function leads to perturbation of cellular geometry during growth. Here the authors study the interaction between RAB-A5c and MTs and establish that the relatively mild phenotype of interference is in large parts due to compensatory changes in MT arrays. They then provide modeling data that can explain the observed phenotypes by a change in edge-vs-face stiffness. I cannot judge the technicalities and validity of their modeling, but I find their conclusions intuitive from a cell biological perspective. Finally, they elaborate on the previously noted connection between MTs and RAB-A5c localisation and provide very nice co-localisation studies that further establish that MTs are required – and at least partially – explain the localisation of RAB-A5c to edges. In summary, I think this is a very nice study, with great imaging and well-done image analysis. It provides a significant contribution to our understanding of cell shape control and growth in plants.

Other comments

The RAB-A5c[N125I] is a RABA5c variant that allows for inducible, dominant-interference with RABA5c function. I have no problem with this approach in principle, I see its advantages and the authors have previously established its specificity. However, I have a problem with the use of "Loss of RAB-A5c function..." as in the legend title of Figure 1. Moreover, in the age of CRISPR/Cas9 – can the authors at least mention why they don't also use knock-outs? Are they lethal, show no phenotypes? What's their explanation for it? This is only if such alleles are available, I do not think it necessary to produce them for the purpose of this work.

I think the "synergistic" (more than additive) phenotype of RAB-A5c[N125I] and an MT mutant, as well as Oryzalin at sub-maximal doses is very impressive and an important point of the paper. I think it would be nice to integrate the Oryzalin data in the main figure. I find the drug treatment in this context more telling than the mutant (it's a specific drug that allows for an inducible, non-constitutive interference with MTs). Similarly, I wondered whether the authors have ever tested the effects of Isoxaben? This is also a specific drug that interferes with CMF deposition and would nicely complement the *pom2* data. If they have it they should put it in.

*Reviewer #2:*

This is a highly interesting study, which explores different ways to achieve anisotropic growth in higher plants. It has been well established that the microtubule-guided deposition of cellulose plays a central role in growth anisotropy. The authors have uncovered an additional process, involving cell edges and the small GTPase RAB-A5c. A role of cell edges in CMT organisation has been described before, but its precise role remains not well characterized and is often ignored. I therefore think the manuscript is of interest for the readers of *eLife*. There are a number of points that need to be addressed in the revision.

– The authors only seem to consider the CMTs on the outer membrane, whereas the CMTs on inner membranes (e.g. on the L1/L2 interface) are not taken into account at all. This is an important issue as there is evidence that the inner and outer CMTs have different behaviours (e.g. a recent Curr Biol paper by Robinson et al., see also Crowell et al., 2011). In hypocotyls, the outer CMTs are less well aligned, unless mechanically challenged, which might be linked to differences in turgor or wall thickness for example. This aspect should be taken into account, as it might affect the interpretation of the results and the modelling outcomes.

– The quality of the CMT imaging is somewhat disappointing and in certain figures, the microtubules are barely visible. Compare e.g. to numerous articles on CMTs in a wide variety of organs including roots and hypocotyls. It would be good to improve this for a re-submission.

– There seems to be some disagreement in the literature on the precise organisation of CMTs in roots. Some reports claim CMTs are highly aligned in epidermal cells of wild type roots (Sugimoto et al., 2003 as a random example), whereas others (as in this study) claim these arrays are more dynamic and can adopt different arrangements (see also e.g. Panteris et al., 2013). Note that this apparently also depends on the hormonal balance and the environment. Till what extent could this dynamic nature of the CMTs influence the experimental outcome and interpretation?

– Note that the botero mutation in KTN also causes a reduction in cellulose content. Burk et al. for example claim a loss 20% in Arabidopsis stems. I wasn't sure how this would influence the interpretation of the results? This should at least be discussed.

In conclusion, this study sheds further light on the intriguing role of cells edges in growth control.

*Reviewer #3:*

Manuscript by Kirchhelle et al., presents an interesting insight into the regulation of cell growth during lateral root development. Authors combined high-resolution cell imaging, genetics and computational approaches to address how mechanical stress is controlled by microtubules and specific polar Rab GTPases. Experiments are performed with care and modeling description seems appropriate. I have only a few comments to the manuscript that authors should consider before this work can be accepted for publication:

1) Subsection “Increased cell wall anisotropy can counteract edge-mediated 186 cell swelling in silico” paragraph three; It is not clear to me how to fold change in anisotropy related to µ_edge_ parameter. Also, I could not find exact values for µ_edge_. How this parameter was estimated?

2) Subsection “Meristematic lateral root cells exhibit anisotropic growth”: Authors choose a 1.5-fold threshold for a volumetric growth of meristematic cells. Is there any justification for that particular choice?

3) Subsection “Computational Model” paragraph ten: only one or two parameters where estimated the rest is taken from measurements in other organs such as a leaf. How model predictions would change if these additional parameters would vary within, for instance, a two-fold range.

4) I could not find any speculation on how Rab-A5c is delivered to its polar domain. Since Rab-A5c contribution is significant to the regulation of cell growth it would be great to see some discussion on that matter.

---

## [Author Response]

Reviewer #1:The manuscript of Kirchhelle et al. provides further insights into the function of RAB-A5C, a member of the largely extended family of RAB11 homologs in plants. […] It provides a significant contribution to our understanding of cell shape control and growth in plants.Other commentsThe RAB-A5c[N125I] is a RABA5c variant that allows for inducible, dominant-interference with RABA5c function. I have no problem with this approach in principle, I see its advantages and the authors have previously established its specificity. However, I have a problem with the use of "Loss of RAB-A5c function..." as in the legend title of Figure 1. Moreover, in the age of CRISPR/Cas9 – can the authors at least mention why they don't also use knock-outs? Are they lethal, show no phenotypes? What's their explanation for it? This is only if such alleles are available, I do not think it necessary to produce them for the purpose of this work.

Conditional overexpression of dominant-negative small GTPase variants like RAB-A5c[N125I] has two major advantages: firstly, this approach can overcome difficulties with redundancy among gene family members, secondly, the inducible pOp/LhGR expression system used here to drive *RAB-A5c[N125I]* expression allows temporal and dosage control. Our preliminary data indicate that the five RAB-A5 subfamily members can indeed act redundantly, as both the single or double T-DNA insertion knock-out mutants have a wild-type phenotype. By contrast, the severe phenotype of *RPS5a>Dex>RAB-A5c[N125I]* seedlingsgerminated under inducing conditions (Kirchhelle et al., 2016) suggests RAB-A5 function is essential during seedling (and possibly other stages of) development, and complete loss of RAB-A5 function is therefore likely lethal. We are in the process of generating quintuple *rab-a5* KO mutants to confirm this prediction, however we note that to understand the role of RAB-A5c during directional growth, such mutants will likely not prove useful. It is precisely the ability to conditionally disrupt RAB-A5c function that allowed us to dissect the relationship between RAB-A5c and cellulose orientation during directional growth control. While we believe that our previous data showing that RAB-A5c[N125I] acts specifically (noted by reviewer 1) justifies our use of “loss of RAB-A5c function”, we acknowledge reviewer 1’s concern this may be confusing to readers associating loss-of-function exclusively with KO mutants. We have therefore replaced “loss of RAB-A5c function” with “inhibition of RAB-A5c function” in the legend title of Figure 1 and other locations in the text for clarity. We have also included in the text a brief description of the advantages of the dominant-negative protein expression strategy.

I think the "synergistic" (more than additive) phenotype of RAB-A5c[N125I] and an MT mutant, as well as Oryzalin at sub-maximal doses is very impressive and an important point of the paper. I think it would be nice to integrate the Oryzalin data in the main figure. I find the drug treatment in this context more telling than the mutant (it's a specific drug that allows for an inducible, non-constitutive interference with MTs).

As requested, we have integrated the Oryzalin data in the absence/presence of 1µM Dex into Figure 3. We have also retained previous Figure S5 (now Figure 3—figure supplement 1) in its original format to show our data using Taxol as and/or sub-saturation levels of Dex.

Similarly, I wondered whether the authors have ever tested the effects of Isoxaben? This is also a specific drug that interferes with CMF deposition and would nicely complement the pom2 data. If they have it they should put it in.

While we agree Isoxaben treatment is an interesting experiment, we think it is out of context in this manuscript. As reviewer 1 points out, Isoxaben is an inhibitor of CMF deposition – specifically, Isoxaben treatment results in the loss of cellulose synthase complexes from the plasma membrane (Paredez et al., 2006). However, our objective in using the *pom2-4* mutant was to assess the effect of perturbations in *orientation* of CMF deposition rather than *rate* of CMF deposition. While a role of CSI1/POM2 in CSC delivery/recycling has been reported (Lei et al., 2015; Zhu et al., 2018), CSC complexes are still present and active at the plasma membrane in *csi1/pom2* mutants. As CSI1/POM2 mediates the association of CSCs in the plasma membrane with microtubules guiding their orientation, orientation of CMF deposition rather CMF deposition *per se* is perturbed in *csi1/pom2* mutants (Li et al., 2012; Bringmann et al., 2012). Therefore, *csi1/pom2* mutants are considered to phenocopy Oryzalin treatment (Li et al., 2012), but not Isoxaben treatment.

Reviewer #2:[…] – The authors only seem to consider the CMTs on the outer membrane, whereas the CMTs on inner membranes (e.g. on the L1/L2 interface) are not taken into account at all. This is an important issue as there is evidence that the inner and outer CMTs have different behaviours (e.g. a recent Curr Biol paper by Robinson et al., see also Crowell et al., 2011). In hypocotyls, the outer CMTs are less well aligned, unless mechanically challenged, which might be linked to differences in turgor or wall thickness for example. This aspect should be taken into account, as it might affect the interpretation of the results and the modelling outcomes.

We originally did not attempt to quantify microtubule orientation at cell faces other than the outer face for two reasons: (1) our genetic and pharmacological interventions which affect all microtubules within a cell as well as our computational data corroborate our hypothesis that microtubule reorganisation in the presence of RAB-A5cNI is compensatory, and (2) quantification of CMTs at cell faces shared with neighbouring cells is associated with considerable technical difficulties. In particular, the cell wall at the L1/L2 interface in roots is less than 200nm thick (Dyson at al., 2014), making it challenging to resolve CMTs at the inner periclinal face of L1 cells from those at the outer periclinal face in the L2 cells. This is exacerbated by the complex surface topologies of cell faces at the L1/L2 interface in comparison to L1 outer faces. We now provide a new image analysis strategy in which we have segmented epidermal cells in 3D based on the plasma membrane marker YFP:NPSN12, and have then projected a co-expressed RFP:TUB6 marker onto the 3D cell surface (Figure 1, E,F; Figure 1—figure supplement 1C,D). This strategy overcomes the difficulty of complex surface topology as the signal is projected normal to the surface from an even distance. Using this technique, we found that in wild type roots, CMT arrays at the L1/L2 interface were in transverse orientation more often than at the outer face (though not always), and arrays had a relatively low overall anisotropy. In the presence of RAB-A5c[N125I], we observed an increase of CMT anisotropy and transversely oriented arrays at the L1/L2 interface similar to our observations at the outer face. As cell wall thickness at the L1/L2 interface is below the z-resolution limit of CLSM, we cannot exclude a contribution of microtubules from the outer face of cortex cells to our measurements at the L1/L2 interface. We therefore also investigated the effect of changing anisotropy at the inner periclinal and anticlinal faces in our computational model (Figure 2). We found that: (1) introduction of anisotropic CMFs at the inner faces reduced cell swelling in comparison to cases were CMFs were isotropic at all faces, (2) reduction of cell wall stiffness at edges caused cell swelling irrespective of CMF orientation at the inner faces, and (3) introduction of anisotropy at the outer face reduced cell swelling irrespective of CMF orientation at the inner faces. Taken together, our new experimental and computational data support our previous conclusions.

– The quality of the CMT imaging is somewhat disappointing and in certain figures, the microtubules are barely visible. Compare e.g. to numerous articles on CMTs in a wide variety of organs including roots and hypocotyls. It would be good to improve this for a re-submission.

We apologise for the bad quality of the CMT images in the paper, which were in largely due to image compression artefacts in the original submission1 which escaped our notice. We are now submitting the original, uncompressed data and include additional data to show microtubule orientation at high resolution (Figure 1—figure supplement 1).

– There seems to be some disagreement in the literature on the precise organisation of CMTs in roots. Some reports claim CMTs are highly aligned in epidermal cells of wild type roots (Sugimoto et al., 2003, as a random example), whereas others (as in this study) claim these arrays are more dynamic and can adopt different arrangements (see also e.g. Panteris et al., 2013). Note that this apparently also depends on the hormonal balance and the environment. Till what extent could this dynamic nature of the CMTs influence the experimental outcome and interpretation?

We are reporting here that in meristematic cells of wild-type lateral roots, CMT arrays at the outer face are relatively disordered (isotropic) and their net orientation is variable with respect to the long axis of the root across a population, and CMT arrays at the inner face show broadly similar degrees of anisotropy, although net orientation appears to be more often in transverse orientation. As cells enter the transition and elongation zone, CMT arrays become anisotropic in transverse orientation (see e.g. Figure 1—figure supplement 1A). While this result is broadly similar to what had been described e.g. in Panteris et al., 2013, we think it is important to note that the existing literature focusses almost entirely on CMT organisation in primary roots, where, based on YFP:RAB-A5c localisation and the effect of RAB-A5c[N125I] expression, RAB-A5c function appears to be less important. We have speculated that the difference in the role of RAB-A5c requirement reflects developmental differences between primary and lateral roots, which may also impact on microtubule organisation. This notion is supported by our observation that GCP2 localisation differs between lateral and primary roots (Figure 1—figure supplement 3). It is conceivable that CMT/CMF organisation is the dominant mechanism controlling directional growth in primary roots, which may explain the more stringent organisation into transverse arrays described in the literature for primary roots. We have expanded our discussion to include these points.

– Note that the botero mutation in KTN also causes a reduction in cellulose content. Burk et al. for example claim a loss 20% in Arabidopsis stems. I wasn't sure how this would influence the interpretation of the results? This should at least be discussed.

We have amended our manuscript to discuss this point. In brief, in our computational model, we test the effect of varying cellulose content through changing the fibre fraction ϑf (Figure 2—figure supplement 2). We found that reducing cellulose content by half had only a minor effect on overall cell swelling relative to changes in mechanical properties at edges and anisotropy at faces, and increasing anisotropy at cell faces could still counter-act the effect of cell swelling induced by reduction of shear moduli at edges. This observation, together with the fact that Oryzalin and Taxol treatment phenocopies the effect observed in the *erh3-3* mutant lines, indicates that the loss of cellulose anisotropy rather than cellulose content is the primary cause for the observed synergistic phenotype between RAB-A5cNI and *erh3-3*.

In conclusion, this study sheds further light on the intriguing role of cells edges in growth control.Reviewer #3:[…] 1) Subsection “Increased cell wall anisotropy can counteract edge-mediated 186 cell swelling in silico” paragraph three; It is not clear to me how to fold change in anisotropy related to µ_edge_ parameter. Also, I could not find exact values for µ_edge._ How this parameter was estimated?

We currently do not have quantitative experimental data available to estimate the µ_edges_ parameter, and suitable experimental methods (e.g. atomic force microscopy) are technically challenging and beyond the scope of this manuscript. To assess the qualitative effect of changing properties at edges, we emulated our previous approach in the previous 2D linear elastic FE model (Kirchhelle et al., 2016) to vary µ_edges_ by factor 10 in comparison to faces. The apparent shear modulus in our current 3D model is a combination of all shear moduli μ_m_, μ_f_, and μ_c_, so we the same factor 10 to all shear moduli to changed the apparent shear modulus. We have updated to manuscript to clarify this point.

2) Subsection “Meristematic lateral root cells exhibit anisotropic growth”: Authors choose a 1.5-fold threshold for a volumetric growth of meristematic cells. Is there any justification for that particular choice?

The threshold for volumetric growth was chosen based on the assumption that entry into the elongation zone coincided with a rapid increase in growth. To identify the threshold, volume growth was plotted against cell’s relative position along the longitudinal axis of the root (see Figure 5—figure supplement 1), and the point of rapid growth increase was estimated to correspond to approximately 1.5-fold volume increase for the 6h time interval. We added a note and Figure Supplement to clarify this.

3) Subsection “Computational Model” paragraph ten: only one or two parameters where estimated the rest is taken from measurements in other organs such as a leaf. How model predictions would change if these additional parameters would vary within, for instance, a two-fold range.

In the previous version of the manuscript, we tested different values for the fibre fraction ϑf, the crosslinking modulus μ_c_, and fibre fractional anisotropy FA. We are now including additional simulations testing the effect of varying the shear moduli for fibres μ_f_ and matrix μ_m_ and the bulk modulus K (Figure 3—figure supplement 2). We note that changes in these parameters affected cell swelling to various degrees without changing the overall predictions of the model (discussed in the main text and Appendix 1).

4) I could not find any speculation on how Rab-A5c is delivered to its polar domain. Since Rab-A5c contribution is significant to the regulation of cell growth it would be great to see some discussion on that matter.

We have previously reported that RAB-A5c localisation at cell edges is sensitive to both actin and microtubule depolymerising drugs (Kirchhelle et al., 2016), informing our working model that actin is required for RAB-A5c compartment delivery to cell edges, whereas microtubules are involved in anchoring compartments in place. Considering the fact that we are not contributing any new data related to RAB-A5c delivery here, and that the discussion is already lengthy, we believe including further speculation on RAB-A5c delivery would not enhance this manuscript.